# BFM-Zero: A Promptable Behavioral Foundation Model for Humanoid Control Using Unsupervised Reinforcement Learning

**Yitang Li**[1,2,*]   **Zhengyi Luo**[2,*]   **Tonghe Zhang**[2,$]   **Cunxi Dai**[2,$]   **Andrea Tirinzoni**[3]
**Anssi Kanervisto**[3]   **Haoyang Weng**[1,2]   **Kris Kitani**[2]   **Mateusz Guzek**[3]   **Ahmed Touati**[3]
**Alessandro Lazaric**[3]   **Matteo Pirotta**[3,†]   **Guanya Shi**[2,†]

[*]Equal Contribution.   [$]Equal Contribution.   [†]Equal advising.
[1]IIIS, Tsinghua University (Work done during authors' internship at CMU)   [2]Carnegie Mellon University   [3]Meta

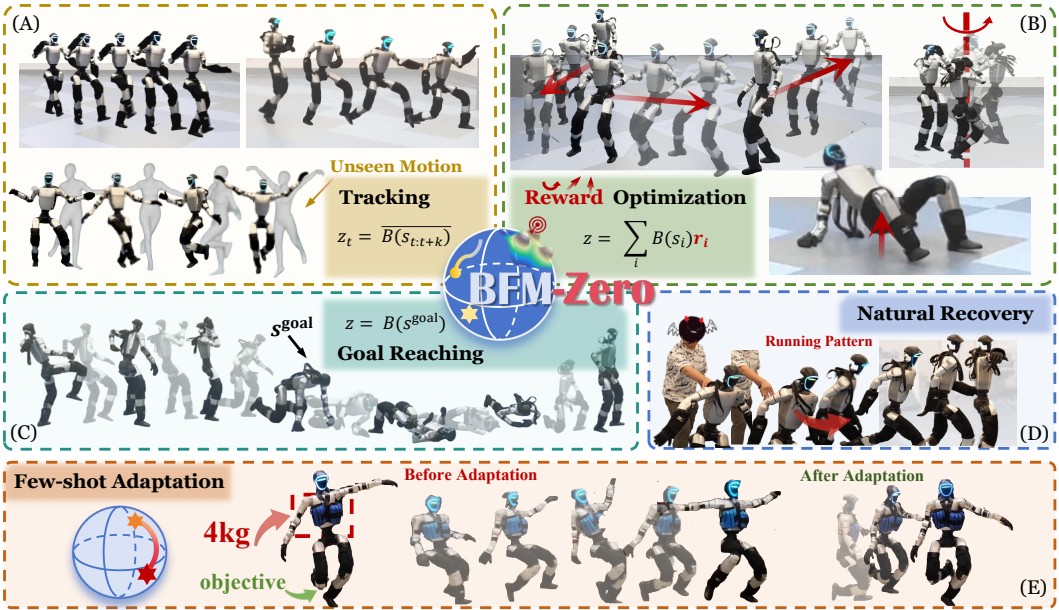

Figure 1: **BFM-Zero** enables versatile and robust whole-body skills. (A-C) Diverse zero-shot inference methods. (D) Natural recovery from large perturbation. (E) Few-shot adaptation.

## Abstract

Building Behavioral Foundation Models (BFMs) for humanoid robots has the potential to unify diverse control tasks under a single, promptable generalist policy. However, existing approaches are either exclusively deployed on simulated humanoid characters, or specialized to specific tasks such as tracking. We propose **BFM-Zero**, a framework that learns an effective shared latent representation that embeds motions, goals, and rewards into a common space, enabling a single policy to be prompted for multiple downstream tasks without retraining. This well-structured latent space in **BFM-Zero** enables versatile and robust whole-body skills on a Unitree G1 humanoid in the real world, via diverse inference methods, including zero-shot motion tracking, goal reaching, and reward inference, and few-shot optimization-based adaptation. Unlike prior on-policy reinforcement learning (RL) frameworks, **BFM-Zero** builds upon recent advancements in unsupervised RL and Forward-Backward (FB) models, which offer an objective-centric, explainable, and smooth latent representation of whole-body motions. We further extend **BFM-Zero** with critical reward shaping, domain randomization, and history-dependent asymmetric learning to bridge the sim-to-real gap. Those key design choices are quantitatively ablated in simulation. A first-of-its-kind model, **BFM-Zero** establishes a step toward scalable, prompt-

able behavioral foundation models for whole-body humanoid control. Webpage: https://lecar-lab.github.io/BFM-Zero/

# 1 INTRODUCTION

Humanoid robots have the potential to transform numerous aspects of our daily lives, from manufacturing and logistics to healthcare and personal assistance. However, realizing this potential requires robots to perform a wide range of tasks in dynamic and unstructured environments. Humanoid whole-body control is a fundamental and challenging problem in robotics, serving as the first step to enable the humanoids to work safely in human environments (Gu et al., 2025).

In robotics, foundation models have the potential to unify diverse control objectives under a single policy, allowing robots to adapt to new tasks in a zero-shot[1] way or with efficient post-training. The closest approaches to such paradigms are Vision-Language-Action (VLA) models for robotic manipulations (e.g., Ghosh et al., 2024; Intelligence et al., 2025; Kim et al., 2024; Zhong et al., 2025; Team et al., 2025; Bjorck et al., 2025) that learn from human demonstrations (i.e., behavior cloning). However, for humanoid whole-body control, there is a fundamental mismatch that limits direct behavior cloning: unlike manipulation tasks, there are no readily available actuator-level action labels or large-scale teleoperation datasets.

For whole-body humanoid control, most recent advancements follow the sim-to-real pipeline and rely on reinforcement learning (RL) to train policies in simulation before transferring them to hardware (Gu et al., 2025). Following the success of RL-based motion tracking in physics-based character animation (e.g., Luo et al., 2024; Tessler et al., 2024; Tirinzoni et al., 2025), recent works (e.g., Zakka et al., 2025; Seo et al., 2025; Chen et al., 2025; Liao et al., 2025; He et al., 2025a; Cheng et al., 2024; He et al., 2025b) have shown remarkable results in transferring policies trained in simulation to real robots. However, most of these approaches rely on *on-policy policy gradient* methods (e.g., PPO (Schulman et al., 2017)) with *explicit tracking-based rewards* and suffer from major limitations. First, they remain task-specific: most policies are trained to explicitly imitate motion capture clips or solve a single task. Second, they are non-adaptive: once trained, policies cannot be easily fine-tuned or composed for new tasks. Third, they lack a unified and explainable interface for goal specification and behavior composition, making it difficult for human operators to direct the robot or combine learned skills into new behaviors.

In this work, we investigate whether *off-policy unsupervised* RL can be a suitable approach to train so-called Behavioral Foundation Models (BFMs) for whole-body control of a humanoid robot, enabling it to solve a wide range of downstream tasks specified by rewards, goals, or demonstrations without retraining. For tasks that require retraining, the BFM should enable efficient post-training. This conjecture is far from trivial. First, most existing methods with real-world deployment rely on on-policy training (primarily PPO), and there is little evidence that off-policy learning—commonly used in unsupervised RL for training multi-task policies—is well suited to this context. Second, no evidence exists that unsupervised RL algorithms can handle the sim-to-real gap and dynamic disturbances robustly, either during simulation policy training or at real-world inference.

We develop **BFM-Zero**[2], an online off-policy unsupervised RL algorithm that leverages motion capture data to regularize the process of learning generalist whole-body control policies towards *human behaviors*. We introduce domain randomization to address the sim-to-real gap and train robust policies via asymmetric history-dependent training, leveraging the privileged information available in simulation. Additionally, we incorporate auxiliary rewards to ensure that the learned behaviors adhere to the safety and operational constraints of the physical robot. To the best of our knowledge, the resulting algorithm allows us to train the *first behavioral foundation model* for real humanoids that can be prompted for different tasks (e.g., reward optimization, pose reaching, and motion tracking) without retraining (i.e., in zero-shot). Such a flexible and ready-to-use model, paves the way to fast adaptation, fine-tuning or even high-level planning. We validate our approach in both simulated environments and on a real Unitree G1 humanoid (Fig. 1 for examples), demonstrating

---

[1]*Zero-shot* means that, after pre-training, the policy can be directly deployed in the real world without further interacting with either simulated or real environments. In contrast, *few-shot* means the policy needs to interact with the environment to collect new data in few episodes to improve on certain tasks.

[2]**Zero** comes from its zero-shot inference capability via unsupervised RL and it is a first-of-its-kind model.

robust generalization across tasks and conditions, and showing that even when the zero-shot policy is not satisfactory, we can effectively improve it in a matter of a few episodes interacting with the environment. Related work discussion is available in Section A.

## 2 BFM-ZERO FOR HUMANOID WHOLE-BODY CONTROL

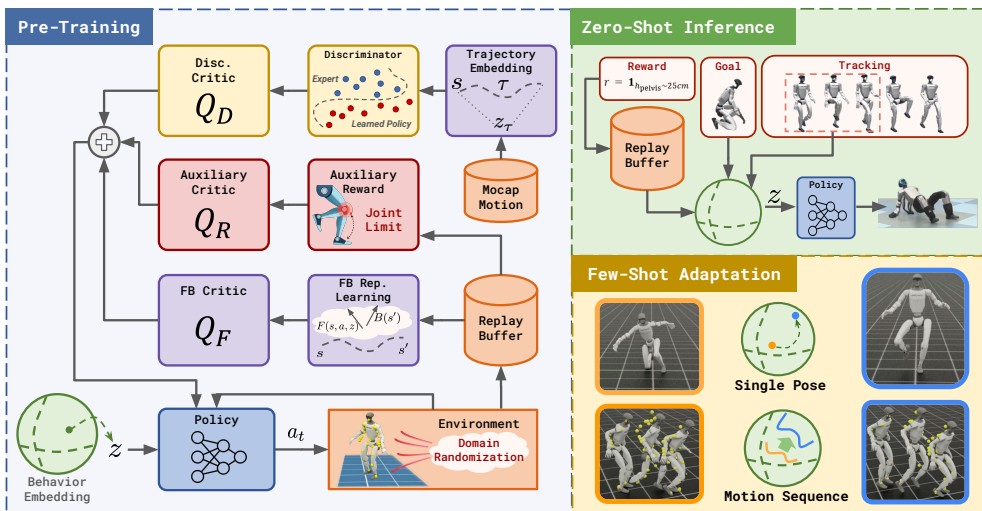

Figure 2: An overview of the **BFM-Zero** framework. After the pre-training stage, **BFM-Zero** forms a latent space that can be used for zero-shot reward optimization, single-frame goal reaching, and tracking. It can also be adapted in a few-shot fashion to reach more challenging poses.

In this section, we outline the pipeline for training **BFM-Zero** in simulation and transferring it to real humanoids. Unlike for virtual characters (e.g., Peng et al., 2022; Tessler et al., 2023; Tirinzoni et al., 2025), applying unsupervised RL to real humanoids has not yet been attempted. Our **BFM-Zero** framework consists of an unsupervised pre-training stage, a zero-shot inference procedure, and possibly a fast-adaptation post-training stage (as shown in Fig. 2). Section 2.1 provides an overview of unsupervised RL using the forward-backward representation framework adopted by **BFM-Zero**. Section 2.2 details **BFM-Zero** pre-training, whose objective is to learn *a unified latent representation* that embeds tasks (e.g., target motions, rewards, goals) into a shared space $Z \subseteq \mathbb{R}^d$ and *a promptable policy* that conditions on this representation to perform diverse behaviors without task-specific retraining. Then, for downstream tasks during inference, we embed the task into the latent space and use the policy to execute the task in a zero-shot manner. We also show that we can efficiently adapt the zero-shot policy in the latent space $Z$ to improve performance on unseen tasks that are not easily covered by zero-shot inference via sampling-based optimization.

**Problem formulation.** We formulate real-world humanoid control as a partially observable Markov decision process (POMDP) defined by the tuple $(S, O, A, P, \gamma)$, where $S$ is the full state space, $O$ is the observation space, $A$ is the action space, $P(s_{t+1}|s_t, a_t)$ is the transition dynamics, and $\gamma \in (0, 1)$ is the discount factor. For the 29-degree-of-freedom (DoF) humanoid, the action $a \in A \subset \mathbb{R}^{29}$ contains the proportional derivative (PD) controller targets for all DoFs. The privileged information ($s \in \mathbb{R}^{463}$) consists of root height, body pose, body rotation, and linear and angular velocities. The observable state $o_t = \{q_t - \bar{q}, \dot{q}_t, \omega_t^{\text{root}}/4, g_t\} \in \mathbb{R}^{64}$ is defined as joint position $q_t \in \mathbb{R}^{29}$ normalized w.r.t. the nominal position $\bar{q}$, joint velocity $\dot{q}_t \in \mathbb{R}^{29}$, root angular velocity $\omega_t^{\text{root}} \in \mathbb{R}^3$ and root projected gravity $g_t \in \mathbb{R}^3$. We denote by $o_{t,H} = \{o_{t-H}, a_{t-H}, \dots, o_t\} \in \mathbb{R}^{93 \cdot H + 64}$ the observable history composed by proprioceptive state and action. All the components of the states (except root height) are normalized w.r.t. the current facing direction and root position. At pre-trainig, we assume that the agent has access to a dataset of unlabeled motions $\mathcal{M} = \{\tau\}$, which contains observation and privileged states trajectories *i.e* $\tau = (o_1, s_1, \dots, o_{l(\tau)}, s_{l(\tau)})$.

## 2.1 UNSUPERVISED RL WITH FORWARD-BACKWARD REPRESENTATIONS

During the pretraining phase, **BFM-Zero** learns a compact representation of the environment by observing online reward-free interactions in the simulator and leveraging an offline dataset of unlabeled behaviors, resulting in a model that can be prompted to tackle a wide range of downstream tasks (e.g., tracking or reward maximization) in a zero-shot manner. To achieve this, we build on top of the recent FB-CPR algorithm (Tirinzoni et al., 2025) which combines the Forward-Backward (FB) method for zero-shot RL (Touati & Ollivier, 2021) with online training and policy regularization on motion-capture data. This method falls in the broader category of unsupervised RL based on successor features (e.g., Touati & Ollivier, 2021; Touati et al., 2023; Pirotta et al., 2024; Park et al., 2024; Agarwal et al., 2024), which involves three components: (i) a latent task feature $\phi : S \to \mathbb{R}^d$ that embeds observation $s \in S$ into a $d$-dimensional vector, (ii) a policy $\pi_z : S \to A$ conditioned on a latent vector $z \in \mathbb{R}^d$, and (iii) latent-conditioned successor features (Barreto et al., 2017) $F_z$ that encode the expected discounted sum of latent task features under the corresponding policy $\pi_z$, i.e, $F_z \simeq \mathbb{E}[\sum_t \gamma^t \phi(s_t) \mid \pi_z]$. We now explain how FB-CPR trains those components.

**FB representations and FB-CPR.** Among the different unsupervised RL approaches, forward-backward (FB) representations provide a principled unsupervised training objective for jointly learning latent task representations and their associated successor features. At a high level, FB learns a finite-rank approximation of long-term policy dynamics, where $\boldsymbol{B}$ captures the low-frequency features that best summarize the long-range temporal dependencies between states. Formally, given a training state distribution $\rho$, the FB framework learns two mappings: a forward mapping $\boldsymbol{F} : S \times A \times \mathbb{R}^d \to \mathbb{R}^d$ and a backward mapping $\boldsymbol{B} : S \to \mathbb{R}^d$ such that the long-term transition dynamics induced by the policy $\pi_z$ decompose as:

$$M^{\pi_z}(\mathrm{d}s' \mid s, a) \simeq \boldsymbol{F}(s, a, z)^\top \boldsymbol{B}(s') \rho(\mathrm{d}s') \tag{1}$$

where for any region $X \subset S$ of the state space, $M^{\pi_z}(s' \in X \mid s, a) := \sum_t \gamma^t \Pr(s_t \in X \mid s, a, \pi_z)$ denotes the discounted visitation probabilities of reaching $X$ under the policy $\pi_z$, starting from the state-action pair $(s, a)$. Eq. 1 implies that $\boldsymbol{F}$ is the successor features of $\phi(s) := (\mathbb{E}_\rho[\boldsymbol{B}(s)\boldsymbol{B}(s)^\top])^{-1}\boldsymbol{B}(s)$ (Touati et al., 2023). The learned representation $\phi$ defines a latent task space by inducing a family of linear reward functions of the form, i.e., $r_z(s) = \phi(s)^\top z$, In particular, each policy $\pi_z$ is optimized to maximize $\mathbb{E}_\rho[\sum_t \gamma^t \phi(s_t)^\top z \mid \pi_z] = \boldsymbol{F}(s, a, z)^\top z$, i.e., $\boldsymbol{F}(s, a, z)^\top z$ is a Q-value function of $\pi_z$ with reward $r = \phi^\top z$. Intuitively, $z \in Z$ defines a *task-centric* latent space associated with the task feature $\phi$, where for each $z$, the corresponding $\pi_z$ optimizes the linear combination of $\phi$, $r_z = \phi^\top z$. As shown in Section 3.4, the $Z$ space learned by **BFM-Zero** is smooth and semantic, and it enables both zero-shot inference and few-shot adaptation. Importantly, in contrast to standard RL approaches, the set of reward functions of interest $\{r_z\}$ is not given (e.g., motion tracking) but learned, and it can represent a wide range of tasks. FB-CPR (Tirinzoni et al., 2025) extends the general FB framework by introducing a latent-conditioned discriminator to regularize the unsupervised learning process to produce policies that are close to a set of demonstrated behaviors in a motion dataset $\mathcal{M}$. Furthermore, while FB algorithm is offline, FB-CPR is trained fully online and off-policy and does not require a full-coverage offline dataset.

## 2.2 BFM-ZERO PRE-TRAINING FOR HUMANOID CONTROL

Before proceeding with the description of implementation details, we identify several design choices that are crucial for achieving sim-to-real transfer in unsupervised RL.

*A) Asymmetric Training.* To bridge the gap between simulation (full state) and real robot (partial observability), we train the policy on observation history $o_{t,H}$, while critics have access to privileged information $(o_{t,H}, s_t)$. This setup improves policy robustness under limited sensing while leveraging privileged critics to provide accurate value estimates. Using history narrows the information gap between proprioceptive actors and privileged critics and improves adaptability under DR.

*B) Scaling up to Massively Parallel Environments.* Inspired by recent work on large-batch off-policy RL (Seo et al., 2025), we scale training across thousands of environments with large replay buffers and high update-to-data (UTD) ratios. This enables efficient unsupervised training of a diverse family of policies while retaining stability, a crucial step for scaling humanoid pretraining.

*C) Domain Randomization (DR).* To enhance robustness and adaptability, we randomize key physical parameters (link masses, friction coefficients, joint offsets, torso center-of-mass) and apply per-

turbations and sensor noise. This prevents overfitting to simulation dynamics and ensures that policies remain stable when deployed on real hardware (see Fig. 4 in Appendix).

*D) Reward Regularization.* In robotics (e.g., He et al., 2025a; Zakka et al., 2025), it is common to incorporate reward regularization techniques to avoid undesirable behaviors. For example, reaching the limit of the joint may lead to highly nonlinear behaviors that are difficult to model in simulation or even damage the robot's hardware.

We train `BFM-Zero` within an off-policy actor-critic scheme. The policy-conditional, *history-based*, *privileged* forward map $\boldsymbol{F}$ and *privileged* backward map $\boldsymbol{B}$ are trained to minimize the temporal difference loss derived from the Bellman equation for successor measures (Touati & Ollivier, 2021). Let $\mathcal{D}$ the replay buffer of online interactions with the simulator and $\nu$ is an arbitrary distribution over $Z$, we consider the following FB objective:

$$\mathcal{L}(\boldsymbol{F}, \boldsymbol{B}) = \mathbb{E}\left[\left(\boldsymbol{F}(o_{t,H}, s_t, a_t, z)^\top \boldsymbol{B}(o^+, s^+) - \gamma\overline{\boldsymbol{F}}(o_{t+1,H}, s_{t+1}, a_{t+1}, z)^\top \overline{\boldsymbol{B}}(o^+, s^+)\right)^2\right]$$
$$- 2\mathbb{E}\left[\boldsymbol{F}(o_{t,H}, s_t, a_t, z)^\top \boldsymbol{B}(o_{t+1}, s_{t+1})\right],$$

where $z \sim \nu$, $(o_{t,H}, s_t, a_t, o_{t+1,H}, s_{t+1}) \sim \mathcal{D}$, $a_{t+1} = \pi(o_{t+1,H}, z)$ and $(o^+, s^+) \sim \mathcal{D}$. $\overline{\boldsymbol{F}}$ and $\overline{\boldsymbol{B}}$ denote the stop-gradient operator.

The auxiliary *history-based*, *privileged* critic $\boldsymbol{Q_R}$ that imposes safety and physical feasibility constraints by incorporating $N_{\text{aux}}$ penalty rewards is learned with a standard Bellman residual loss:

$$\mathcal{L}(\boldsymbol{Q_R}) = \mathbb{E}_{\substack{(o_{t,H}, s_t, a_t, s_{t+1}) \sim \mathcal{D} \\ z \sim \nu, a_{t+1} = \pi(o_{t+1,H}, z)}}\left[\left(\boldsymbol{Q_R}(o_{t,H}, s_t, a_t, z) - \sum_{k=1}^{N_{\text{aux}}} r_k(s_t) - \gamma\overline{\boldsymbol{Q_R}}(o_{t+1,H}, s_{t+1}, a_{t+1}, z)\right)^2\right].$$

Finally, we employ the *history-based*, *privileged* discriminator critic $\boldsymbol{Q_D}$ that grounds the unsupervised training toward human-like behaviors by assigning rewards based on a latent-conditioned discriminator. This acts both as a style regularization as well as a bias in the online exploration process. As in (Tirinzoni et al., 2025), we employ a variational representation of the Jensen-Shannon divergence and train the discriminator $\boldsymbol{D}$ with a GAN-style objective:

$$\mathcal{L}(\boldsymbol{D}) = -\mathbb{E}_{\tau \sim \mathcal{M}, (o,s) \sim \tau}\left[\log(\boldsymbol{D}(o, s, z_\tau))\right] - \mathbb{E}_{(o,s,z) \sim \mathcal{D}}\left[\log(1 - \boldsymbol{D}(o, s, z))\right].$$

where $z_\tau = \frac{1}{l(\tau)}\sum_{(o,s) \in \tau} \boldsymbol{B}(o, s)$ is a zero-shot imitation embedding of the motion $\tau$. We can then fit a *style* critic $\boldsymbol{Q_D}$ with a Bellman residual loss similar to the auxiliary critic with a reward $r_d(o_t, s_t, z) = \frac{\boldsymbol{D}(o_t, s_t, z)}{1 - \boldsymbol{D}(o_t, s_t, z)}$. Bringing together these critiques results in the final actor loss.

$$\mathcal{L}(\pi) = -\mathbb{E}_{\substack{(o_{t,H}, s_t) \sim \mathcal{D} \\ a_t = \pi(o_{t,H}, z), z \sim \nu,}}\left[\boldsymbol{F}(o_{t,H}, s_t, a_t, z)^\top z + \lambda_D \boldsymbol{Q_D}(o_{t,H}, s_t, a_t, z) + \lambda_R \boldsymbol{Q_R}(o_{t,H}, s_t, a_t, z)\right].$$

**Zero-shot inference.** At test time, `BFM-Zero` can be used to solve different tasks in *zero-shot* fashion, i.e., without performing additional task-specific learning, planning, or fine-tuning. Given an *arbitrary* reward function $r(s)$, the corresponding Q function of $\pi_z$ can be formulated as

$$Q_r^{\pi_z}(s, a) = \int_{s'} M^{\pi_z}(\mathrm{d}s'|s, a) r(s') \simeq \mathbb{E}_{s' \sim \rho}[\boldsymbol{F}(s, a, z)^\top \boldsymbol{B}(s') r(s')] = \boldsymbol{F}(s, a, z)^\top \mathbb{E}_{s' \sim \rho}[\boldsymbol{B}(s') r(s')].$$

Since $\boldsymbol{F}(s, a, z)^\top z$ is the Q function of $\pi_z$, we have $z_r = \mathbb{E}_{s' \sim \rho}[\boldsymbol{B}(s) r(s)]$. In practice, we can leverage a sample-based estimate, given by $z_r = \frac{1}{N}\sum_i r(s_i)\boldsymbol{B}(s_i)$ where $s_i \in \mathcal{D}$ and $\mathcal{D} = \{(s_i, r_i)\}$ is obtained by subsampling the online replay buffer. For a goal-reaching task, we have $z_g = B(s_g)$. Finally, for **tracking** a motion $\tau = \{s_1, \ldots, s_n\}$, a sequence of policies $\{z_t\}$ is obtained as $z_t = \sum_{t'=t}^{t+H} \boldsymbol{B}(s_{t'})$, where $H$ is a look-ahead horizon (Pirotta et al., 2024).

**Few-Shot Adaptation.** We can leverage optimization techniques for adaptation in latent space $Z$ using online interaction with the simulator at test time. We demonstrate this by refining a static pose or an entire motion to maximize $J(z) = \sum_{t=0}^{T-1}\left(r_{\text{task}}(s_t) - \alpha_R \sum_{k=1}^{N_{\text{aux}}} r_k(o_t, s_t, a_t)\right)$. For **single-pose adaptation**, we use the zero-shot policy $z_0 = B(s_g, o_g)$ as initial point and apply the Cross-Entropy Method (CEM) (Rubinstein, 1999; Rubinstein & Kroese, 2004). For **trajectory-level adaptation**, we warm-start from a tracked motion sequence and perform zero-order, sampling-based trajectory optimization over a *sequence* of latent prompts, $\mathbf{z}_{t:t+H-1}$, using a dual-loop annealing schedule in the spirit of DIAL-MPC (Xue et al., 2025). This procedure consistently stabilizes challenging segments and reduces motion-tracking error, while retaining the human-like prior given by the discriminator without finetuning networks.

| Model | Test env. | Test data | Track | Rwd | Pose |
|---|---|---|---|---|---|
| **BFM-Zero**-*priv* | Isaac (no DR) | LAFAN1 | 1.0749 | 299.3 | 1.0291 |
| **BFM-Zero** | Isaac (DR) | LAFAN1 | 1.1015 | 221.9 | 1.1387 |
| **BFM-Zero** | Mujoco (DR) | LAFAN1 | 1.0789 | 207.3 | 1.1041 |
| **BFM-Zero** | Mujoco (DR) | AMASS | 1.0342 | 207.3 | 1.4735 |

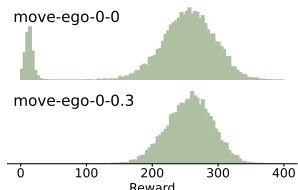

Figure 3: Tracking, reward, and goal-reaching performance across models for different testing configurations (left), and example distributions of reward evaluation scores for **BFM-Zero** in Isaac (DR) (right). Each metric is averaged over tasks. We consider the average return over episodes lasting 500 steps for reward, the average joint position error $E_{\mathrm{mpjpe}}$ averaged over the whole motion for tracking, and the error $E_{\mathrm{mpjpe}}$ averaged over the episode for goal-reaching.

## 3 EXPERIMENTS

In this section, we thoroughly evaluate **BFM-Zero** both in simulation and in real. We train **BFM-Zero** in a simulated version of Unitree G1 using IsaacLab (Mittal et al., 2023) at 200 Hz, while the control frequency is 50 Hz. For the behavior dataset, we use the LAFAN1 dataset (Harvey et al., 2020) retargeted to the Unitree G1 robot. The LAFAN1 dataset contains 40 several-minute-long motions. We also demonstrate generality of **BFM-Zero** on a Booster T1 humanoid (App. **??**).

### 3.1 ZERO-SHOT VALIDATION IN SIMULATION

In this section, we quantitatively assess the performance and robustness of **BFM-Zero** along different dimensions in simulation.

**Asymmetric learning and domain randomization.** We consider a *privileged* version of **BFM-Zero** where all components of the algorithm receive privileged information. We train this model in a simulated environment with nominal dynamical parameters (*No DR*) and we test it in the very same configuration. This serves as an idealized configuration similar to the problems where unsupervised RL was previously shown to work (Tirinzoni et al., 2025), although it leads to a model that is *not deployable* on the real robot. We then compare to **BFM-Zero** trained and tested on a domain randomized version of the environment (*Sim DR*), which corresponds to the model actually deployed on the real robot. Overall, **BFM-Zero** is $2.47\%$, $25.86\%$, $10.65\%$ worse than **BFM-Zero**-priv across tracking, reward, and pose reaching tasks. This shows that despite the algorithmic changes made in **BFM-Zero** compared to FB-CPR, the learning dynamics is still correct and the model retains a satisfactory performance compared to its idealized version. Interestingly, reward tasks suffer from a larger drop in performance. This is in part due to the sparse nature of the reward functions we consider, which makes them less forgiving to suboptimal behaviors and amplify any model error. We also conjecture that this may be related to the reward inference process with domain randomized data. In Fig. 3 we also show the distribution of the performance of **BFM-Zero** for two representative reward functions across repetitions of the inference process[3] and episodes. While for `move-ego-0.3` the performance is fairly consistent, for `move-ego-0.0`, we notice that a few instances obtained a very poor performance. We conjecture that this is related to the increased randomness of the data observed during training due to domain randomization, which makes inference with a small subsampled dataset more brittle and prone to failure.

**Sim-to-sim performance.** We evaluate the robustness of **BFM-Zero** to the dynamics of the humanoid by testing it in Mujoco. We notice that performance difference is limited (i.e., all variations are less than 7%), showing that the domain randomization at training and the history components in the actor and critics contribute to a good level of robustness and adaptivity.

**Out-of-distribution tasks.** Finally, we evaluate **BFM-Zero** on a different set of tracking and pose reaching tasks obtained from the AMASS dataset (Mahmood et al., 2019). We consider 175 out-of-distribution motions from the CMU subset of the AMASS and 10 manually-selected poses from the motions in the entire AMASS dataset. We run tests in Mujoco to combine different dynamics and

---

[3]In the reward inference, we use a dataset of states randomly subsampled from the training dataset. As a result, multiple repetitions of the process may return different policies.

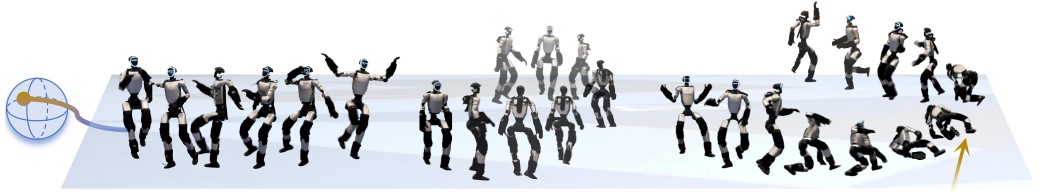

Figure 4: Real-World Validation of **Tracking**. *Left:* Highly dynamic dancing. *Middle:* Frequently turning during walking. *Right:* **Naturally** recover to continue track the motion.

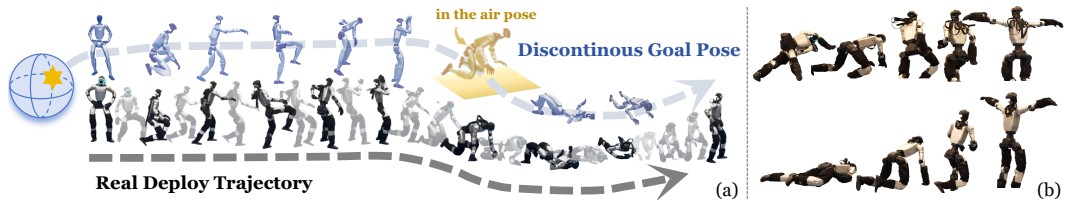

Figure 5: Real-World Validation of **Goal Reaching**. (a) Continuously goal-reaching: the blue/yellow pose denotes the goal pose, while black marks the real robot pose, and gray visualizes the transition between each pose. (b) Transition from any pose to T-pose.

out-of-distribution tasks. While a direct comparison of performance between LAFAN1 and AMASS tasks may be misleading due to the specific nature of the motions and poses used in the evaluation, we notice that overall `BFM-Zero` is able to successfully generalize and complete tracking and pose reaching even when exposed to tasks that are not represented in the training data.

## 3.2 ZERO-SHOT REAL-WORLD VALIDATION

Finally, we deploy the `BFM-Zero` model zero-shot on a real Unitree G1 robot. In real-world validation, we aim to **1)** qualitatively confirm the model's tracking, reward optimization, and goal reaching capabilities on a few selected tasks; **2)** assess its robustness to perturbations and failures (e.g., falling). *All results in this section come from **one** model.*

**Tracking** As shown in Fig. 1, 4, `BFM-Zero` enables the robot to track diverse motions, including various walking styles, highly dynamic dances, fighting, and sports. Even when unstable or falling (*Right*), it demonstrates remarkably gentle, natural, and safe behavior while recovering and continues tracking seamlessly. This capability stems not merely from robustness gained through disturbance training, but mostly from *TD-based off-policy training* and the use of a GAN-based reward which explicitly encourages human-likeness and regularization terms that enable it to draw upon a rich skill library—much like a human—to adaptively complete the tracking task. Additionally, to evaluate the coverage and generalization capability, we used real videos and retargeted them to the G1. Despite the suboptimal motion quality and discontinuities introduced by occlusions of monocular videos and artifacts in video estimation, the system is robust to lower quality data and can still successfully track these motions.

**Goal Reaching** For the goal-reaching task, we extract a sequence of target poses by randomly sampling goal states and discarding their velocity components. The zero-shot latent of these poses are then permuted and sequentially provided to the policy. As illustrated in Fig. 5, the robot consistently converges to a natural configuration that closely approximates the target pose, even when the target is infeasible (the Yellow one in Fig. 5). Moreover, the resulting trajectory exhibits smooth and natural transitions, whether between successive targets(5.a) or from an arbitrary pose to the T-pose(5.b), without the need for explicit interpolation, demonstrating the smoothness and high coverage of the learned skill space.

**Reward Optimization** We evaluate reward optimization in the real world with three task families: (i) locomotion rewards that specify base velocities and angular velocities, (ii) arm-movement rewards that command wrist height, and (iii) pelvis-height rewards that request sitting, crouching,

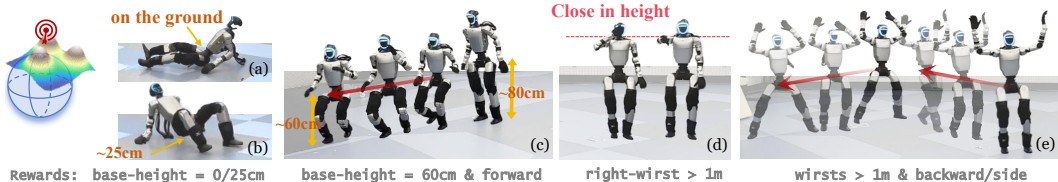

Figure 6: Real-World **Reward Optimization**. The red arrow represents the base velocity tracking target. (a) `sitting`; (b) `crouch-0.25`; (c) `move-low0.6-ego-0-0.7`; (d) Diverse behaviors from *one* reward `raisearm-m-l`; (e) combing `raisearm-m-l` with `move-ego-180-0.3` and `move-ego--90-0.7`.

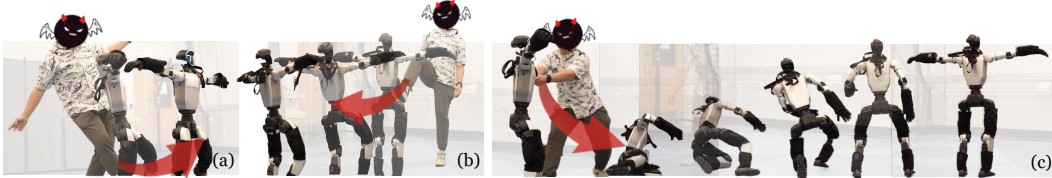

Figure 7: Disturbance Rejection: (a) Keeps steady when kicked in the leg. (b) Absorbs a hard push with one smooth rear step. (c) *Naturally* stands up and returns to T-pose after being yanked down.

or low-movement (Fig. 6(a–c)); reward definitions in Appendix C. With simple reward definitions, the robot faithfully executes base-height, base-velocity, and arm-movement commands. Composite skills can be derived from simply linear combination of the rewards (e.g., going backward while raising arms), demonstrating controllable, skill-level interpolability. Also, given a specific reward, averaging over different mini-batches from the replay buffer yields a set of latent variables that represents a diverse collection of potential optimal modes. (Fig. 6(d)). Formulating objectives through reward functions makes our policy intuitive for human users and receptive to language prompts.

**Disturbance Rejection** One notable advantage of our policy is its strong compliance and robustness. As illustrated in Fig. 1 and 7, our framework enables the robot to withstand severe disturbances—such as fierce pushes, kicks, or even being dragged to the ground, while recovering in a natural, human-like manner. For instance, after a strong forward shove, the robot instinctively closes its arms, takes several rapid steps in a running-like pose, and then gradually slows down before reopening its arms (Fig. 1). This level of robustness goes beyond the typical demonstrations seen in previous works: rather than fiercely reacting to the disturbances, our policy autonomously adapts. Although it receives only a single latent vector from the static T-pose as input, it can automatically deviate from the reference posture, adopt a dynamic recovery pose, and eventually return to tracking the original T-pose just as a human would.

### 3.3 EFFICIENT ADAPTATION FOR **BFM-ZERO**

In this section we show how we leverage adaptation to improve the zero-shot inference performance.

**Single Pose Adaptation** We validate *few-shot single-pose adaptation* on hardware with an additional **4 kg** mass rigidly attached to the torso link. Starting from the zero-shot latent $z^{\text{init}}$, we apply CEM to obtain $z^{\star}$, augmenting the rollout objective with a sparse task term $r = \mathbf{1}_{\{h_{\text{right foot}} > 0.15 \text{ m} \wedge \text{no-contact}\}}$, which encourages right-foot clearance while avoiding unintended contacts. As shown in Fig. 8 (a), without adaptation, the motion driven by $z^{\text{init}}$ destabilizes and produces an environmental collision within 5 s. In contrast, the optimized prompt $z^{\star}$ maintains single-leg balance for over 15 s. These results indicate that prompt-level optimization alone can compensate for the payload-induced dynamics shift, without retuning network parameters.

**Trajectory Adaptation** For trajectory adaptation, we focus on optimizing a leaping motion under altered ground friction. We perform dual-annealing trajectory optimization (Xue et al., 2025) using the explicit tracking reward defined in (Luo et al., 2023). We used sampling with particle count $N = 2048$, temperature schedules $\beta_1 = 0.85$ and $\beta_2 = 0.9$, and optimization iterations $M = 6$. The reward curve and before/after adaptation key-point tracking performance is shown in Fig. 8(b), showing that our method significantly improves tracking accuracy, reducing error by ∼*29.1%*.

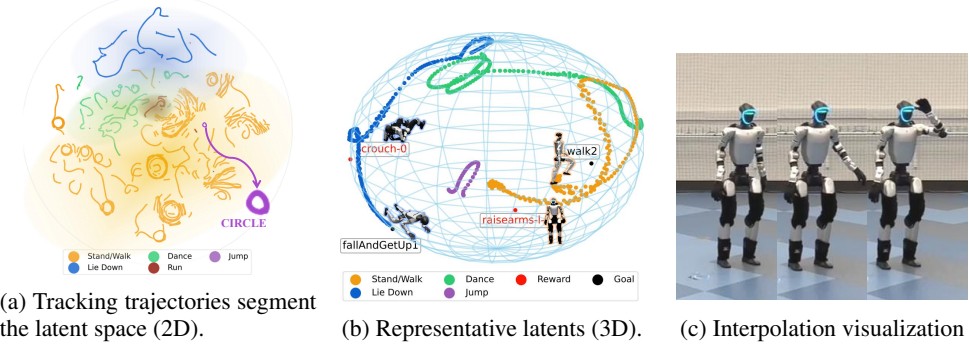

Figure 8: Few-Shot Adaptation: (a) Single-pose adaptation improving single-leg standing under an additional payload. (b) Trajectory adaptation reduces tracking error.

(a) Tracking trajectories segment the latent space (2D).

(b) Representative latents (3D).

(c) Interpolation visualization

Figure 9: Latent space visualization and analysis.

## 3.4 THE LATENT SPACE STRUCTURE OF BFM-ZERO

As mentioned in Sect. 2.1, **BFM-Zero** provides an **interpretable** and **structured** representation of the behaviors of a humanoid robot. This representation not only facilitates understanding of the policy space but also enables instantaneous interpolation of existing skills without retraining.

**Visualizing the Latent Space** To examine the structure of the latent space, we sample latent vector trajectories and project them onto a two-dimensional plane (Fig. 9a) to visualize the space, and also use a three-dimensional sphere to present representative latent generated for *tracking, reward optimization and goal reaching*(Fig. 9b) using t-SNE (van der Maaten & Hinton, 2008). We can see the latent space is organized by motion style: semantically similar trajectories cluster, revealing a shared task centric structure.

**Motion Interpolation on the Latent Space** The structured nature of $\mathcal{Z}$ enables smooth interpolation between latent representations. We can leverage Spherical Linear Interpolation (Jafari & Molaei, 2014) to generate intermediate latent vectors along the geodesic arc between the two end-points. To evaluate interpolated behaviors, we feed the resulting in-between $z_{t=0.5}$ into the **BFM-Zero** policy, and deploy it on both simulated and real humanoid robots. As shown in Fig. 9c, the interpolated policy produces *semantically meaningful* intermediate skills in a *zero-shot* manner. These behaviors compose immediately—*no additional training* required.

## 4 DISCUSSION

In this paper we showed for the first time that off-policy unsupervised RL is a viable approach to train a behavioral foundation model for whole-body control of a real humanoid robot. While **BFM-Zero** shows a remarkable level of generalization and robustness, it still suffers from several limitations: **1)** The scope and performance of the behaviors expressed by **BFM-Zero** is connected to the motions used in training. Investigating the connection between the size of motion datasets, simulated datasets, architecture and model performance (e.g., quantity and quality of the learned behaviors) and consolidating it into scaling laws is important to guide future iterations of this approach. **2)** While history-based actor and critics and domain randomization reduced the sim-to-real

gap, we believe algorithms with better online adaptation capabilities are needed to reliably express more complex movements. **3)** While we performed a preliminary investigation of test-time adaptation, a more thorough understanding of fast adaptation and fine-tuning of these models is needed to broaden their practical applicability.

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

# Appendix

## A  RELATED WORK

In recent years, learning-based methods have made significant progress in whole-body control for humanoid robots. The largest body of work has focused on simulated humanoids. While these methods have demonstrated impressive capabilities in generating complex and dynamic behaviors using reinforcement learning (Peng et al., 2018; Luo et al., 2023; 2024; Tessler et al., 2024; Serifi et al., 2024), sim-to-real transfer remains a critical challenge in deploying learned policies on real-world humanoid robots. Various strategies have been proposed to bridge this gap, including domain randomization, system identification, asymmetric training, etc. However, the majority of these methods focus on single-task learning, where a policy is trained to perform a specific task, such as walking, running and get up (Radosavovic et al., 2024a;b; Chen et al., 2024; Seo et al., 2025; Zakka et al., 2025; He et al., 2025c; Peng et al., 2017).

Recently, mostly 2025, there has been a surge of interest in developing multi-task and generalist humanoid control policies that can perform a wide range of tasks (He et al., 2024b; 2025a; Zhang et al., 2025; Zeng et al., 2025; Yin et al., 2025; Chen et al., 2025; He et al., 2024a; 2025b; Cheng et al., 2024; Dugar et al., 2025; Fu et al., 2024). The majority of these methods builds on top of approaches developed for simulated humanoids, and enhance them to be robust enough for sim-to-real transfer. While ASAP (He et al., 2025a) pre-train motion tracking policies in simulation and deploy them on the real robot to collect data to train a delta (residual) action model, the most common approach is to first train a motion tracking policy (or multiple policies) in simulation, and then distill it into a single multi-task policy that can perform all the skills in the motion dataset. Common approaches for distillation include using a conditional variational autoencoder to learn a latent space of skills and doing online distillation (He et al., 2024b; Yin et al., 2025; Zeng et al., 2025; Chen et al., 2025; Zhang et al., 2025) or using diffusion models (Liao et al., 2025). However, all these methods require two stages of training to enable promptable policies, they are inherently limited by the quality of the motion since the base policies are trained to track the motion, and they relay on on-policy RL algorithms. Our method represents a significant departure from this paradigm

by directly learning a promptable multi-task policy using an off-policy RL algorithm, which offer a much more reach and structured space of skills, and is not limited by the quality of the motion dataset.

## B    TRAINING DETAILS

The agent interacts with the environment via episodes of fix length $T = 500$ steps. The algorithm has access to the dataset $\mathcal{M}$ containing observation-only motions. Similarly to (Tirinzoni et al., 2025), the initial state distribution of an episode is a mixture between randomly generated falling positions and states in $\mathcal{M}$ (motion initialization). We use prioritization to sample motions from $\mathcal{M}$ and, inside a motion, the state is uniformly sampled. We use an exponential prioritization scheme based on the agent's ability to track a motion. To have a more fine-grained prioritization, we split the 40 LAFAN1 (Harvey et al., 2020) motions into chunks of 10 seconds. Every $N_{\text{eval}}$ interaction steps, we evaluate all the motions and update the priorities base on the earth mover's distance (Rubner et al., 2000, EMD). For each motion $m \in \mathcal{M}$, the priority is given by

$$p(m) \propto 2^{\max\left\{0.5;\, \min\left\{\text{EMD}(m), 2\right\}\right\} \cdot 4}$$

We take inspiration from the recipe in FastTD3 (Seo et al., 2025) to scale up unsupervised off-policy RL to using massively parallel environments. We use standard MLPs for all the components of the model, even for handling history. We simulate $N_{\text{env}}$ parallel (and independent) environments at each step. We scale the buffer size accordingly to the number of environments, following the rule $N_{\text{buffer}} \times N_{\text{env}} \times T$. We use a batch size of $N_{\text{batch}}$ and we use an update-to-data ratio of $N_{\text{ups}}$ gradient steps per (parallel) environment step. We train the model for a total number of environment steps $N_{\text{train}} = \frac{N_{\text{grad}} N_{\text{env}}}{N_{\text{ups}}}$. We report the value of these parameters in Tab. **??**, the missing parameters are as in (Tirinzoni et al., 2025).

**Network architectures.** We use a residual architecture for actor and critics ($F$, $Q_D$) with blocks akin to those of transformer architectures (Vaswani et al., 2017), involving residual connections, layer normalization, and Mish activation functions (Misra, 2020). We use a simple MLP for the critic $Q_R$. We use an ensemble composed of two networks for critics. For discriminator and backward map we use a standard MLP with ReLu activation. Refer to Tab. 3 for more details.

### B.1    **BFM-ZERO**

We provide here a sketch of **BFM-Zero** in (Alg. 1). We report the algorithm without parallel networks for clarity. for clarity as well, we report the FB loss here. Let $a_i' \sim \pi(x_i', z_i)$ where $x_i = (o_{i,H}, s_i)$, then

| Parameter | Value |
|---|---|
| **Environment and Training Setup** | |
| History Length $H$ | 4 |
| Episode Length $T$ | 500 |
| $N_{\text{env}}$ | 1024 |
| $N_{\text{batch}}$ | 1024 |
| $N_{\text{ups}}$ | 16 |
| $N_{\text{grad}}$ | 3M |
| $N_{\text{train}}$ | $\approx$ 192M |
| $N_{\text{buffer}}$ | 10 |
| $N_{\text{eval}}$ | $N_{\text{train}}/20$ |
| Buffer Size (transitions) | $\approx$ 5M |
| Discount Factor | 0.98 |
| Number of Seeding Steps | $10 \cdot N_{\text{env}}$ |
| Fall Initialization Probability | 0.3 |
| **Learning and Regularization** | |
| Sequence Length (Trajectory Sampling) | 8 |
| Latent Dimension $d$ | 256 |
| Discriminator Reg. Coef. $\alpha_D$ | 0.05 |
| Reward Reg. Coef. $\alpha_R$ | 0.02 |
| Gradient Penalty | 10 |
| Learning Rate $F$ | $3 \cdot 10^{-4}$ |
| Learning Rate $B$ | $10^{-5}$ |
| Learning Rate $D$ | $10^{-5}$ |
| Learning Rate Actor $\pi$ | $3 \cdot 10^{-4}$ |
| Learning Rate $Q_D$ | $3 \cdot 10^{-4}$ |
| Learning Rate $Q_R$ | $3 \cdot 10^{-4}$ |
| Orthonormality Loss Coefficient | 100 |
| **Inference** | |
| Number of samples for reward inference | 400000 |
| Tracking look ahead in sim | Seq. length |
| Tracking look ahead in real | 3 (real) |

Table 1: Train parameters

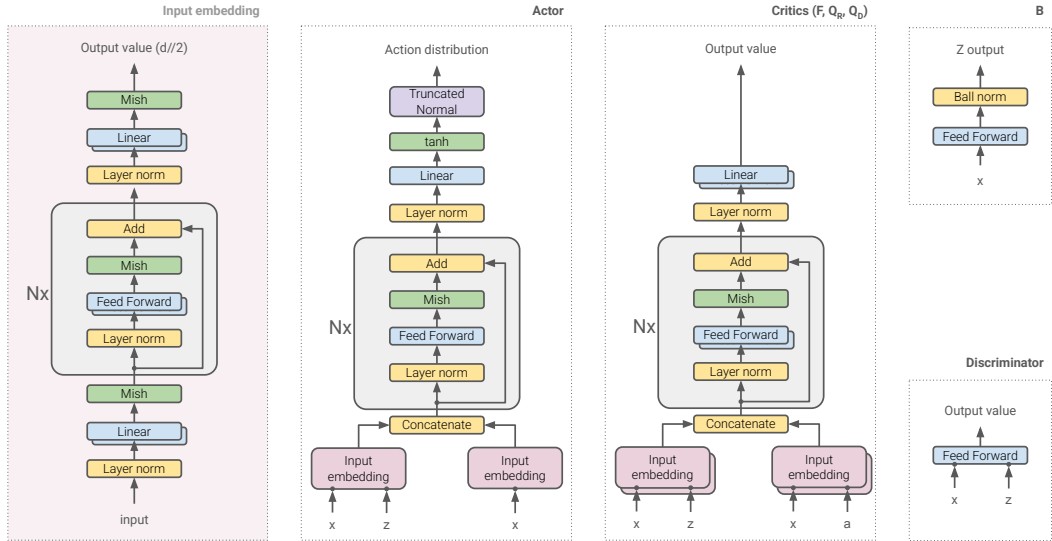

Table 2: Visual representation of the network architectures.

| Hyperparameter | Critics ($F$, $Q_D$, $Q_R$) | Actor | Discriminator | B |
|---|---|---|---|---|
| Input Variables | $(x, a, z)$ | $(x, z)$ | $(x, z)$ | $(x)$ |
| Output Dim | F: $d$, $Q_D$, $Q_R$: 1 | 29 | 1 | $d$ |
| Observation Variable $x$ | $(o_{t,H}, s_t)$ | $o_{t,H}$ | $(s_t, o_t)$ | $(s_t, o_t)$ |
| Embedding Residual Blocks | 4 | 4 | – | – |
| Embedding Hidden Units | 2048 | 2048 | – | – |
| Residual Blocks | 6 | 6 | – | – |
| Feed Forward Hidden Layers | 1 | 1 | 2 | 1 |
| Feed Forward Hidden Units | 2048 | 2048 | 1024 | 256 |
| Activations | Mish | Mish | ReLU | ReLU |
| Number of Parallel Networks | 2 | 1 | 1 | 1 |
| Num. Parameters (no target) | F: 135.8M, $Q_D$, $Q_R$: 134.8M | 31.9M | 2.9M | 0.2M |
| **Total Parameters** | **440.5M** | | | |

Table 3: Network architecture parameters used for real tests. $s_t$ is the privileged information and $o_t$ is the proprioceptive information. $o_{t,H} = \{o_{t-H}, a_{t-H}, \ldots, o_t\}$ denotes the history of proprioceptive states and actions.

| **Domain Randomization** | | | **Additive Observation Noise** | | | **Regularization Rewards** | |
|---|---|---|---|---|---|---|---|
| Parameter | Range | | Observation | Range | | Name | Weight |
| COM Offset [m] | $\mathcal{U}([-0.02, 0.02])$ | | $q_t - \bar{q}$ | $\mathcal{U}([-0.01, 0.01])$ | | DoF Limit | $-10$ |
| Link Mass | $\mathcal{U}([0.95, 1.05])$ | | $\dot{q}_t$ | $\mathcal{U}([-0.5, 0.5])$ | | Action Rate | $-0.1$ |
| Friction | $\mathcal{U}([-0.5, 1.25])$ | | $\text{grav}_t$ | $\mathcal{U}([-0.05, 0.05])$ | | Self Contact | $-1$ |
| Default Joint Pos [m] | $\mathcal{U}([-0.02, 0.02])$ | | $\dot{\omega}_t^{\text{root}}/4$ | $\mathcal{U}([-0.05, 0.05])$ | | Feet Orientation | $-0.4$ |
| Push Robots [m/s] | $\mathcal{U}([0, 0.5])$ | | | | | Ankle Roll | $-4$ |
| | | | | | | Feet Slip | $-2$ |

Table 4: Additional training configurations.

$$\ell_{\text{fb}} = \frac{1}{2n(n-1)} \sum_{i \neq k} \left( F(x_i, a_i, z_i)^\top B(s'_k, o'_k) - \gamma \overline{F}(x'_i, a'_i, z_i)^\top \overline{B}(s'_k, o'_k) \right)^2$$

$$- \frac{1}{n} \sum_i F(x_i, a_i, z_i)^\top B(o'_i, s'_i)$$

$$+ \frac{1}{2n(n-1)} \sum_{i \neq k} \left( B(s'_i, o'_i)^\top B(s'_k, o'_k) \right)^2 - \frac{1}{n} \sum_{i \in [n]} B(s'_i, o'_i)^\top B(s'_i, o'_i) \tag{2}$$

$$+ \frac{1}{n} \sum_{i \in [n]} \left( F(x_i, a_i, z_i)^\top z_i - \overline{B}(s'_i, o'_i) \Sigma_{\overline{B}} z_i - \gamma \overline{F}(x'_i, a'_i, z_i)^\top z_i \right)^2$$

---

**Algorithm 1** **BFM-Zero** Pre-Training

---

1: Initialize empty train buffer: $\mathcal{D}_{\text{online}} \leftarrow \emptyset$
2: Initialize expert buffer $\mathcal{M}$ with action-free trajectories
3: **for** $t = 1, \ldots$ **do**
4:     *//Online interaction*
5:     Sample $\boldsymbol{z}_t = \{z_e\}_{e=1}^{N_{\text{env}}} \in \mathbb{R}^{N_{\text{env}} \times d}$ (if needed)
6:     Execute $\boldsymbol{a}_t \sim \pi(\boldsymbol{o}_{t,H}, \boldsymbol{z}_t) \in \mathbb{R}^{N_{\text{env}} \times A}$ in the *simulated* environments
7:     Store $(\boldsymbol{s}_t, \boldsymbol{o}'_{t,H}, \boldsymbol{a}_t, \boldsymbol{s}'_t, \boldsymbol{o}'_{t+1,H}, \boldsymbol{z}_t)$ in $\mathcal{D}_{\text{online}}$
8:     *//Update*
9:     **for** $j = 1, \ldots, N_{\text{ups}}$ **do**
10:         Sample a batch of $n = N_{\text{batch}}$ transitions $\{(o_{i,H}, s_i, a_i, o'_{i,H}, s'_i, z_i)\}_{i=1}^n$ from $\mathcal{D}_{\text{online}}$
11:         Sample a batch of $\frac{n}{T_{\text{seq}}}$ sequences $\{(w_{j,1}, w_{j,2} \ldots, w_{j,T_{\text{seq}}})\}_{j=1}^{\frac{n}{T_{\text{seq}}}}$ from $\mathcal{M}$ where $w = (s_t, o_t)$
12:         *//Encode expert and update discriminator*
13:         $z_j \leftarrow \frac{1}{T_{\text{seq}}} \sum_{t=1}^{T_{\text{seq}}} B(w_{j,t}) \,;\, z_j \leftarrow \sqrt{d} \frac{z_j}{\|z_j\|_2}$
14:         $\ell_{\text{discriminator}} = -\frac{1}{n} \sum_{j=1}^{\frac{n}{T_{\text{seq}}}} \sum_{t=1}^{T_{\text{seq}}} \log D(w_{j,t}, z_j) - \frac{1}{n} \sum_{i=1}^n \log(1 - D(s_i, o_i, z_i))$
15:         *//Update representation F and B so that $F(s, a; z)^\top B(s') \approx M^{\pi_z}(ds'|s, a)$*
16:         Refer to Eq. 2
17:         *//note that D does not use history*
18:         Compute discriminator reward: $r_i^D \leftarrow \log(D(s_i, o_i, z_i)) - \log(1 - D(s_i, o_i, z_i)), \quad \forall i \in [n]$
19:         Let $x_i = (o_{i,H}, s_i)$ and sample $u_i \sim \pi(o_{i,H}, z_i)$ for all $i \in [n]$. Then
20:         $\ell_{\text{critic}_D} = \frac{1}{n} \sum_{i \in [n]} \left( Q_D(x_i, a_i, z_i) - r_i^D - \gamma \overline{Q_D}(x'_i, a_i, z_i) \right)^2$
21:         $\ell_{\text{critic}_R} = \frac{1}{n} \sum_{i \in [n]} \left( Q_R(x_i, a_i, z_i) - \sum_k r_k^{\text{aux}}(x'_i) - \gamma \overline{Q_R}(x'_i, a_i, z_i) \right)^2$
22:         $\ell_{\text{actor}} = -\frac{1}{n} \sum_{i \in [n]} \left( F(x_i, u_i, z_i)^\top z_i + \alpha_D Q_D(x_i, u_i, z_i) + \alpha_R Q_R(x_i, u_i, z_i) \right)$
23:         *//Update target networks*
24:     **end for**
25: **end for**

---

## C   Tasks and Metrics

In this section we provide a complete description of the tasks and metrics.

**Goal-based evaluation**   We have manually extracted 21 "stable" poses (i.e., states with zero velocities) from the train dataset (i.e., LAFAN1) and 10 poses from the test dataset (i.e., AMASS). We report the selected poses from LAFAN1 in Fig 10. To evaluate how close is the agent to the goal pose, we use the joint error defined as following

$$E_{\text{mpjpe}}(e, g) = \frac{1}{|e|} \sum_{t=1}^{|e|} \|q_t(e) - q(g)\|_2$$

where $e$ is an episode and $q$ is the joint position (i.e., 29D). We report the average across goals. Episodes are of fix length $H = 500$.

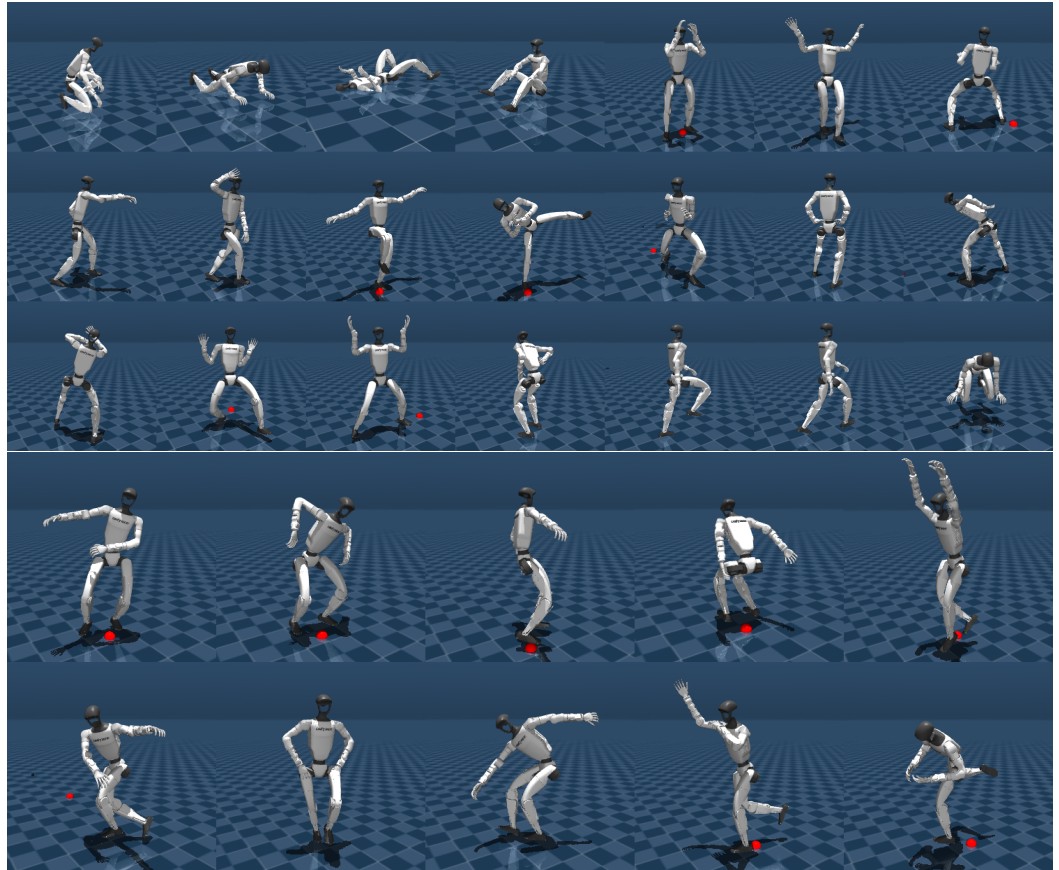

Figure 10: Goal poses selected from frames of the LAFAN1 dataset (Harvey et al., 2020) (top) and AMASS dataset (Mahmood et al., 2019) (bottom).

**Tracking evaluation**   This evaluation aims to assess the ability of the model to imitate a sequence of poses, by ideally matching both positions and velocities. We evaluate the agent both on the train dataset (i.e., LAFAN1) and on out-of-distribution motions selected from AMASS (retargeted to G1). In particular, we randomly selected 175 motions from the CMU dataset of AMASS. For evaluation, we use the same metric as in goal evaluation, i.e.,

$$E_{\mathrm{mpjpe}}(e, m) = \frac{1}{|e|} \sum_{t=1}^{|e|} \|q_t(e) - q_t(m)\|_2$$

and we report the average across motions.

**Reward evaluation**   We define 6 categories of rewards inspired by (Tirinzoni et al., 2025). Rewards are a function of the next state and normalized in $[0, 1]$.

*Standing*.   We evaluate the ability of the agent to stand with the pelvis at different heights. `move-ego-0-0` requires pelvis above 60cm and zero velocity, while `move-ego-low0.5-0-0` requires the pelvis to be between 50cm and 65cm.

*Locomotion*.   This category includes rewards related that requires the agent to move at a certain speed, in a certain direction and at a certain height. We consider 5 representative rewards (`move-ego-0-0.7`, `move-ego-90-0.7`, `move-ego--90-0.7`, `move-ego-0-0.3`, `move-ego-180-0.3`) which include forward, lateral and backward movement. We additionally test also walking forward but with the pelvis at a low height (`move-ego-low0.6-0-0.7`).

*Rotation.* We require the robot to rotate along the vertical axis (i.e., while standing). We consider rotating clockwise and counter clockwise (i.e., `rotate-z-5-0.5` and `rotate-z--5-0.5`).

*Ground poses.* To further stress the ability of the model to control the vertical position, we define rewards requiring the agent to sit on the ground (`sitting`) or having the pelvis slightly above the ground (`crouch-0.25` is about 25cm above the ground).

*Arm raise.* We require the robot to stand in a steady position and to reach certain vertical position with the arms (measured at the wrists). We consider low ($z \in [0.6m, 0.8m]$) and medium ($z > 1m$) positions for the wrists, with soft margins (`raisearms-l-l`, `raisearms-l-m`, `raisearms-m-l`, `raisearms-m-m`).

*Combined rewards.* We finally evaluate the ability of the agent to maximize rewards that require combining multiple skills. In particular, we test combinations of locomotion and rotation with arm movements. We selected 8 combinations of rewards.

Overall, we tested 24 rewards and evaluted perfomance via the cumulative return over episodes of $T = 500$ steps. The initial state of an episode is the default pose.

## D  USE OF LARGE LANGUAGE MODELS

The authors use LLM tools for refining the expression in the final draft of the paper, but did not use it for idea generation, training, and other experiments.

# E  APPLICATION ON BOOSTER T1

We additionally evaluate the generality of our framework by testing **BFM-Zero** on Booster T1 humanoid robot. The LAFAN1 dataset is retargeted to T1 using LocoMujoco (Al-Hafez et al., 2023) and we train the policy with exact same hyper-parameters as G1. The algorithm shows strong generalization ability, allowing T1 also to perform natural walking and expressive dancing motions, as shown in Figure 11a.

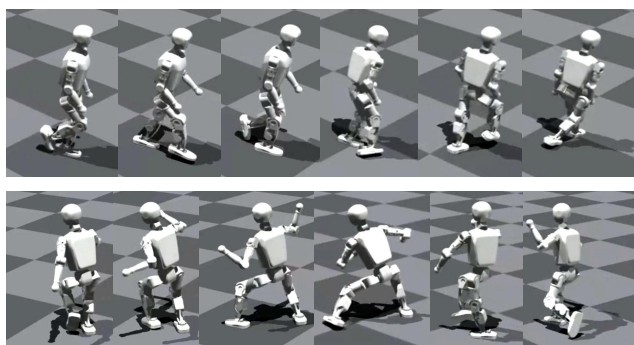

(a) Qualitative results in simulation.

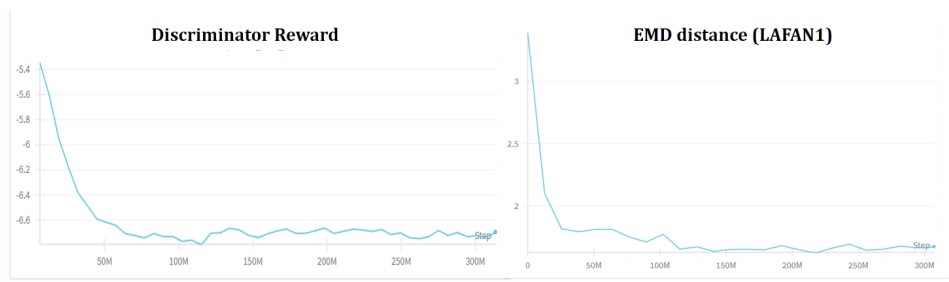

(b) Training Curves.

Figure 11: Applications of **BFM-Zero** on Booster T1.

To demonstrate convergence, we report the training curves (Figure 11b) of the discriminator rewards together with the validation EMD distance over the course of training. The curves show rapid convergence. (Note: lower discriminator rewards indicate that the policy more effectively deceives the discriminator.)

# F  MORE ABLATIONS

## F.1  MODEL AND DATA ABLATIONS FOR TRACKING AND REWARD

We perform ablations on both the data and model size. For training the model in the main paper, we used only the LAFAN1 dataset (Harvey et al., 2020). In these ablations, we additionally leverage motions from the CMU and BMLHandball subsets of AMASS (Mahmood et al., 2019). We selected these two sub-datasets for computation/time reasons and because we believe they contains sufficiently different behaviors than the one in LAFAN1. We consider individual datasets (referred to as LAFAN1 and AMASS in the figures), as well as datasets obtained by merging $X$ percent of the two datasets (with $X = \{12.5\%, 25\%, 50\%, 75\%, 100\%\}$).

We consider different network architectures, including simple feed-forward networks and residual architectures with a varying number of blocks (see Table 6).

We evaluate tracking and reward inferece. For tracking, we use the same test dataset as in (Tirinzoni et al., 2025), but we removed motions from CMU and BMLHandball to ensure complete separation from the training datasets. For reward inference, we use $600,000$ samples from the LAFAN1 motion

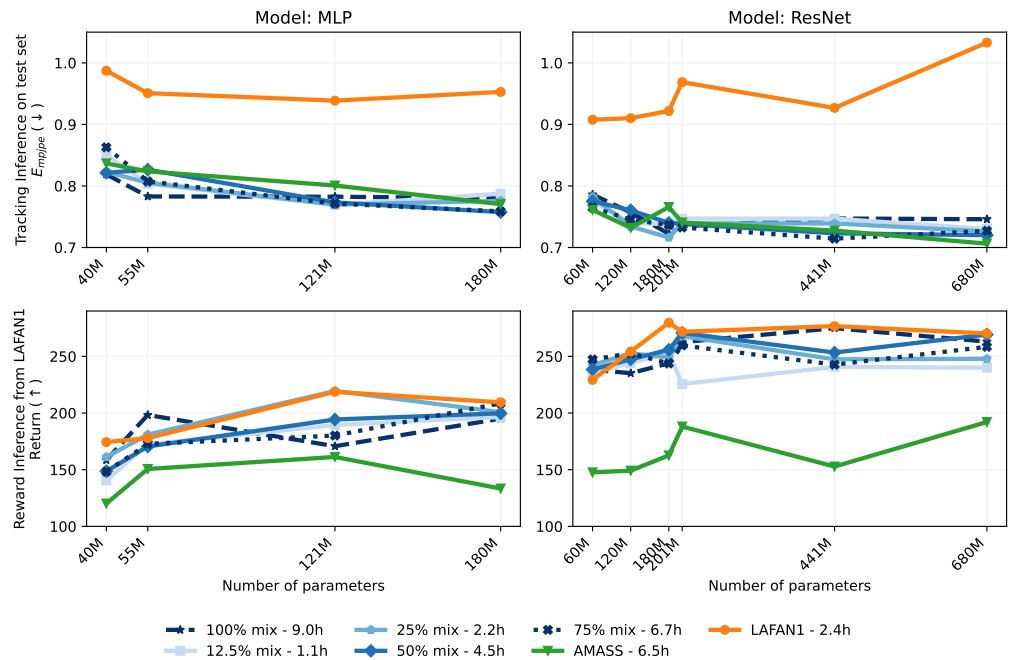

Figure 12: Tracking and reward performance on the test set for different models and datasets. The lower the better for tracking and the higher the better for reward. Reward inference is done using $600,000$ samples from the LAFAN1 expert buffer.

| Dataset | Frames @30FPS | Duration |
|---------|---------------|----------|
| LAFAN1 | 258600 | 2.4h |
| AMASS | 698437 | 6.5h |
| 12.5% | 119620 | 1.1h |
| 25% | 234982 | 2.2h |
| 50% | 475793 | 4.4h |
| 75% | 721720 | 6.7h |
| 100% | 957037 | 8.9h |

Table 5: Number of frames per dataset

dataset and repeated the inference on 10 batches. This is different w.r.t. to what done in the main paper where we used the replay buffer. We refer the reader to the paragraph below for a detailed discussion.

| Architecture | Model | Number of Parameters | | | | | | |
|---|---|---|---|---|---|---|---|---|
| | | $\pi$ | $Q_R$ | $B$ | $Q_D$ | D | F | Total |
| ResNet | 3-block, 2048dim | 19.3M | 59.2M | 201k | 59.2M | 2.9M | 60.3M | 201.1M |
| ResNet* | 6-block, 2048dim | 31.9M | 134.8M | 201k | 134.8M | 2.9M | 135.9M | 440.5M |
| ResNet | 9-block, 2048dim | 44.5M | 210.4M | 201k | 210.4M | 2.9M | 211.5M | 679.9M |
| ResNet | 3-block, 1024dim | 5.5M | 17.0M | 201k | 17.0M | 2.9M | 17.6M | 60.2M |
| ResNet | 6-block, 1024dim | 8.6M | 36.0M | 201k | 36.0M | 2.9M | 36.5M | 120.1M |
| ResNet | 9-block, 1024dim | 11.8M | 54.9M | 201k | 54.9M | 2.9M | 55.4M | 180.1M |
| MLP | 2-layer, 1024dim | 4.4M | 10.7M | 201k | 10.7M | 2.9M | 11.2M | 40.1M |
| MLP | 2-layer, 2048dim | 15.1M | 34.0M | 201k | 34.0M | 2.9M | 35.0M | 121.2M |
| MLP | 4-layer, 1024dim | 6.5M | 14.9M | 201k | 14.9M | 2.9M | 15.4M | 54.8M |
| MLP | 4-layer, 2048dim | 23.5M | 50.8M | 201k | 50.8M | 2.9M | 51.8M | 179.9M |

Table 6: Configurations of the architectures and total number of parameters. $\star$ denotes the configuration used in the main paper.

**Tracking.** We report the results of our ablation in Figure 12 over a single seed. As we increase the total capacity of the model, tracking performance improves for almost all of the mocap datasets used for regularization (see top line). When using LAFAN1 as motion dataset, performance saturates quite early. We believe this is because test motions, and despite being separated from any training motion set, are likely much closer to the motions in CMU and BMLHandball than to those in LAFAN1. We can further notice that residual architectures achieve better performance w.r.t. simple MLP architectures, and we can scale residual architectures to larger sizes. Furthermore, we found training to be instable when scaling MLP to larger architectures (certain seeds diverged).

**Reward.** We observe a mild improvement trend for reward inference when increasing the model size. However, training with LAFAN1 (in some proportion) appears to be important in this case, as reward performance drops when we train only with the subset of AMASS. We also evaluated reward inference performance using both the training buffer (i.e., the replay buffer generated by the agent) and the training motion set that may differ from the LAFAN1 dataset. In both cases, the average performance decreases, with a much more significant drop when using the training buffer. We believe this may be due to the fact that 1) samples in the buffer are collected with domain randomization, whereas the motion buffers are not randomized; 2) motions in LAFAN1 are more aligned with the majority of tasks we evaluate. To validate this conjecture, we report in Figure 14 the distribution of returns for each tasks for a model trained using only LAFAN1 motions (6 block ResNet architecture with 2048 hidden dim) and reward inference done with LAFAN1 or the replay buffer. We can notice that inference with LAFAN1 leads to better policies and thus returns on tasks covered in the dataset (e.g., moving forward, backward, on the side). However, we are able to infer better policies for "non-standard" tasks (e.g., rotate, crouching, move with low height) when using the experience self generated by the agent because these samples are collected by directly deploying policies learned by the agent. Selecting the optimal dataset for reward inference could be an interesting direction for future research.

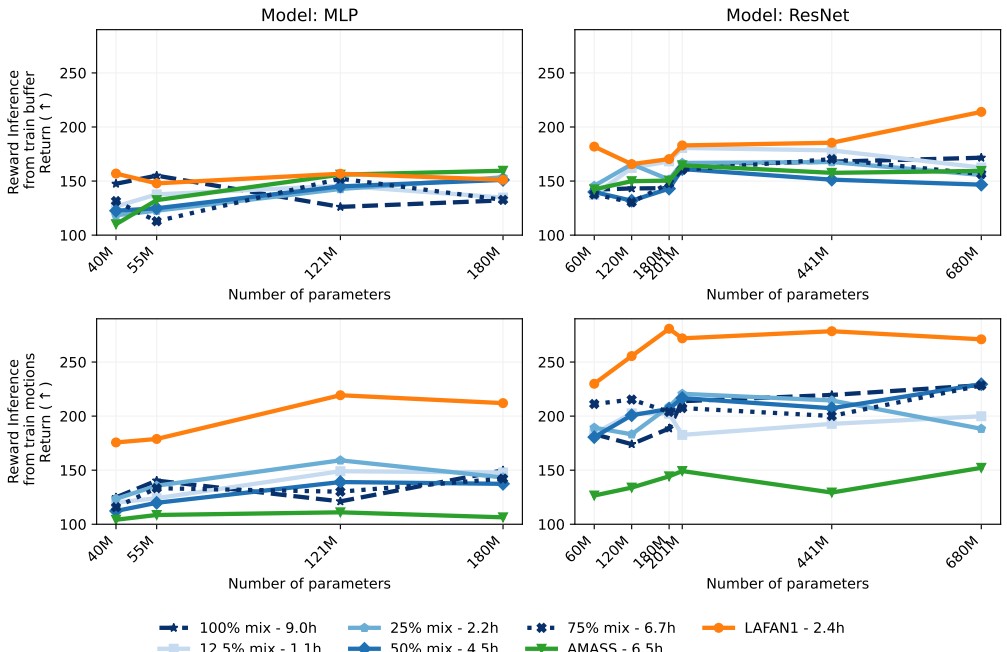

Figure 13: Reward inference performance when using the experience generated by the agent (i.e., online replay buffer) or the motion dataset used for training. We get better reward performance when using the motion dataset, in particular when using LAFAN1 (see Fig. 12).

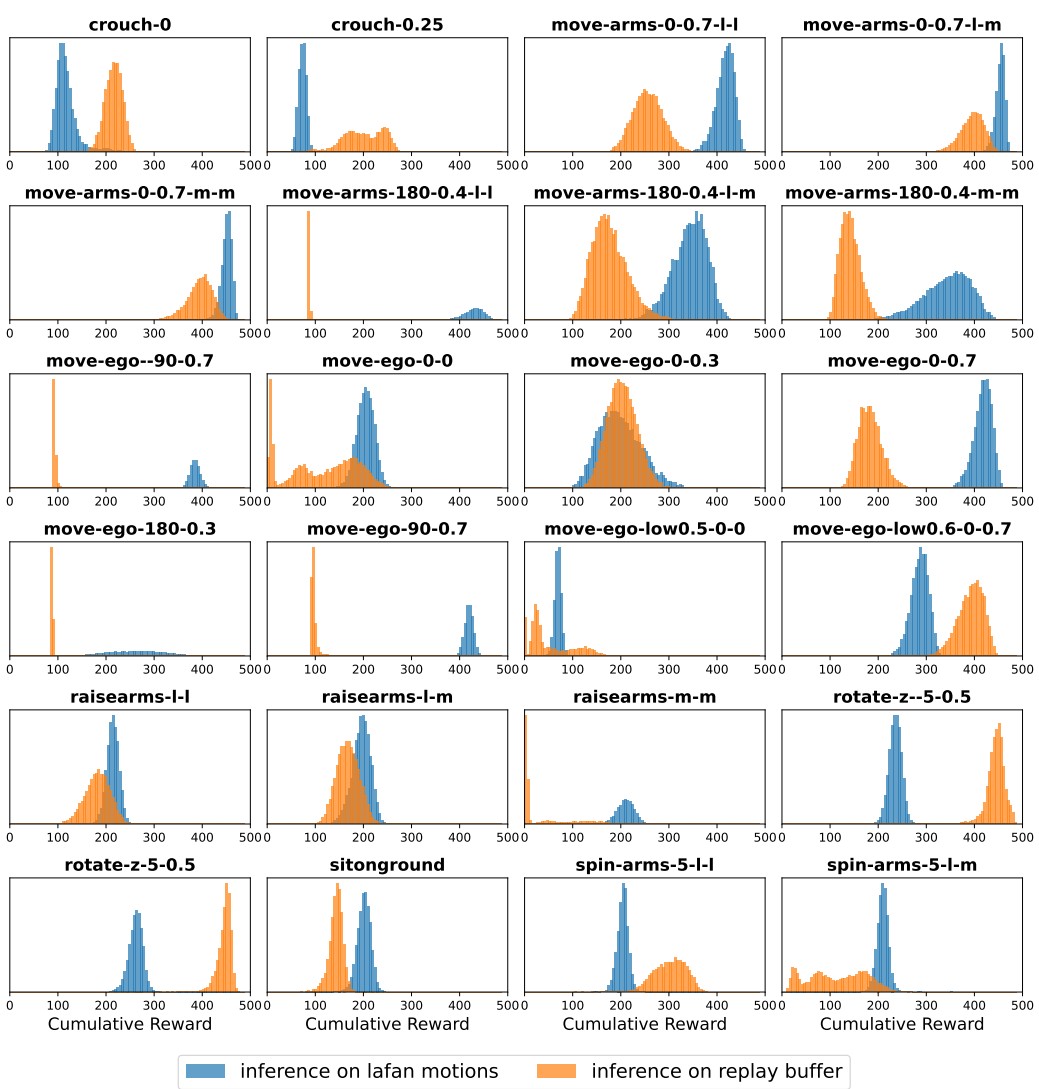

Figure 14: Return distributions for different tasks when using LAFAN1 motions or the agent self-generated samples (replay buffer) for reward inference. Reward inference was repeated 10 times leading to evaluating 10 different policies for each tasks. Each policy was evaluated over 20 episodes.

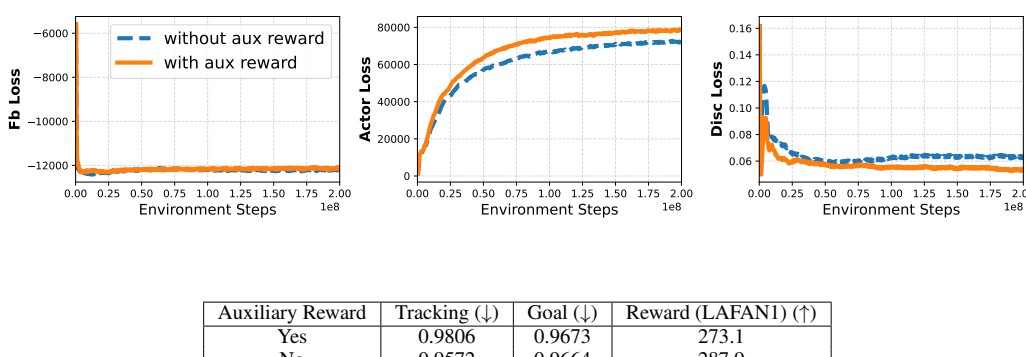

| Auxiliary Reward | Tracking (↓) | Goal (↓) | Reward (LAFAN1) (↑) |
|---|---|---|---|
| Yes | 0.9806 | 0.9673 | 273.1 |
| No | 0.9572 | 0.9664 | 287.9 |

Figure 15: Examples of training losses as a function of the number of environment steps when training with and without auxiliary rewards (top). Bottom table shows the evaluation score of the models on tracking, goal reaching and reward inference. The model is the ResNet model with 6 blocks and 2048 hidden dimension.

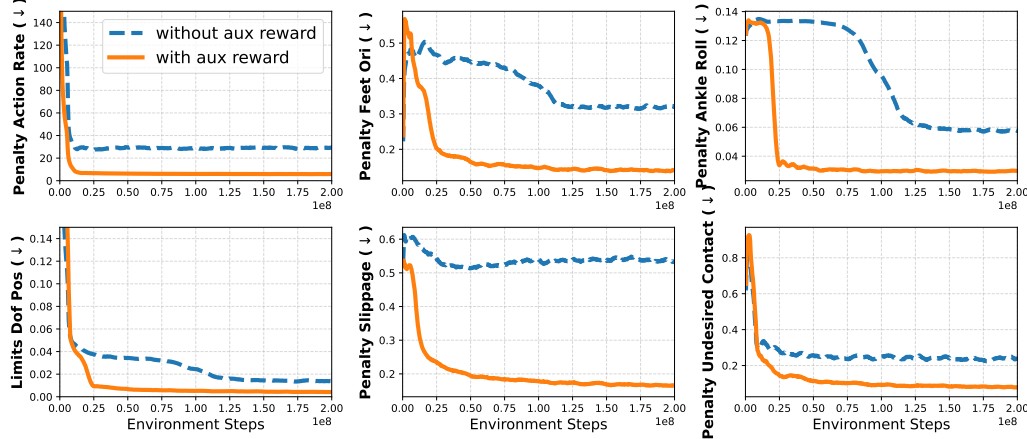

Figure 16: Auxiliary reward metrics as a function of the number of steps in training. We can clearly see that the model trained with auxiliary rewards optimizes the provided auxiliary tasks.

## F.2 DR AND AUX REWARDS

The Domain Randomization (DR) and Reward Regularization (RR) terms are primarily informed by physical constraints and the sim-to-real gap. For example, we penalize large actions because motors exhibit highly nonlinear behavior near joint limits—a phenomenon that the simulator does not fully capture. In simulation, the impact of enabling or disabling these components is difficult to observe, as metrics and training outcomes remain very similar. However, on the real robot, these terms have a significant effect, and their inclusion is essential.

For example, removing auxiliary rewards causes the robot to become highly unstable in the real world, primarily for two reasons: 1) The robot tends to use large control inputs, which result in behaviors that are not accurately modeled by the simulator. 2) Foot placement becomes less precise than in simulation, leading to unstable poses. By inspecting the training curves related to auxiliary penalties, we can clearly see that these metrics (e.g., action rate and feet orientation) are optimized by the model using auxiliary rewards (see Figure 16). In contrast, training curves and evaluation metrics in simulation (see Figure 15) remain very similar regardless of whether auxiliary rewards are included. In some cases, performance in simulation may even appear to improve when auxiliary rewards are removed (see table in Figure 15). However, this improvement does not translate to the real robot. The gap between simulation and reality complicates research iterations, as simulation metrics cannot always be relied upon to predict real-world performance.

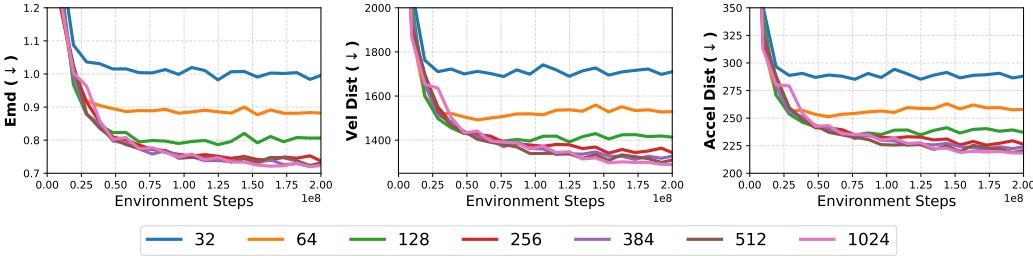

Figure 17: Evaluation curves for tracking during training as a function of the steps in the environment for different latent space dimension (i.e., $d_z$). These metrics are computed on the motion dataset used for training (i.e., LAFAN1). Results are averaged over 2 seeds.

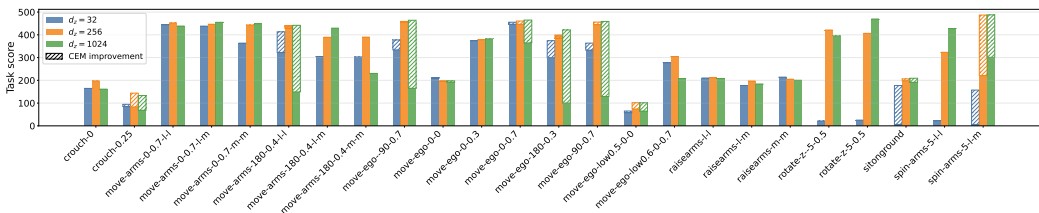

Figure 18: Reward evaluation score split by individual tasks for models with different latent space dimension $d_z$, along with the score increase after CEM adaptation on selected tasks.

### F.3 ABLATION ON Z DIMENSION

In this section, we investigate the impact of the dimensionality of the latent space $Z$ on the model performance. The results in the main paper corresponds to dimension 256, here we ablate the following values $d_z \in \{32, 64, 128, 256, 384, 512, 1024\}$. We kept all the other training parameters fixed as in Table 3 and Table 1.

Figure 17 reports the tracking performance on the train motion dataset as a function of the number of steps in the environment. We can see that low z dimensions (32, 64, 128) lead to poor performance. Models with a dimension of 256 or more have similar performance. When moving to the test evaluation (see table 7), we observe a similar trend for all inference modalities; that is, the performance of models with dimensions of 256 or more is very similar. We do not observe any performance degradation that shows that the training is stable, we even observe an improvement in tracking and reward inference. Only when moving to dimension 1024 we observe a drop in performance in reward inference. This may due to the fact that the zero-shot inference becomes much more complicated in such large space.

To validate this, we adapt the zero-shot latent via CEM optimization on selected tasks, including ones where 1024 latent dimensional models struggled. We run CEM for 20 iterations with population size of 1024, and sweep search space standard deviation over $0.01, 0.1, 0.5, 1.0, 5.0$ and top-elites to keep over $1\%, 2\%, 5\%, 10\%$. We repeat CEM per task three times. Then per task and model, we report the best average CEM result over the parameter sweep. Figure 18 shows that model with z dimension 1024 has poor zero-shot performance in some move ego tasks (e.g., `move-ego-90` and `move-ego-180`), but after CEM adaptation matches or outperforms 32 and 256 dimensional models. This indicates that, indeed, zero-shot inference is more difficult at high dimensional latent space, but the model is still capable of reaching high performance with correct adaptation.

| Z dimension | Tracking ($\downarrow$) | Goal ($\downarrow$) | Reward (LAFAN1) ($\uparrow$) |
|---|---|---|---|
| 32 | 1.3963 | 1.6870 | 213.8224 |
| 64 | 1.2455 | 1.3141 | 240.4211 |
| 128 | 1.0512 | 1.0830 | 263.9501 |
| 256 | 0.9806 | 0.9673 | 273.0826 |
| 384 | 0.9266 | 0.9704 | 290.9922 |
| 512 | 0.9232 | 0.9758 | 307.3982 |
| 1024 | 0.8986 | 0.9875 | 252.3514 |

Table 7: Test metrics for the different inference modalities for models trained with different z dimensions. The base configuration is has the one in the main paper (see Table 3 and Table 1). Results are averaged over 2 seeds.

## G   QUANTITATIVE REAL-WORLD VALIDATION

To further evaluate the model's sim-to-real transferability, we choose several representative motions, measure their real-world tracking errors (Table 8), and compare these results with their simulation counterparts. `sim-nodr` corresponds to a perfectly matched simulator with dynamics centered in the training range, whereas `sim-dr` incorporates dynamic randomization.

| Motions | Start Frame | End Frame | $E_{\mathrm{mpjpe}}^{\mathrm{real}}$ ($\downarrow$) | $E_{\mathrm{mpjpe}}^{\mathtt{sim-nodr}}$ ($\downarrow$) | $E_{\mathrm{mpjpe}}^{\mathtt{sim-dr}}$ ($\downarrow$) |
|---|---|---|---|---|---|
| Dance-1 | 1 | 2400 | 1.1945 | 0.7631 | 0.7736 |
| Dance-2 | 1 | 2000 | 1.0003 | 0.9047 | 0.9235 |
| Walk slowly | 1 | 320 | 0.8662 | 0.5746 | 0.6561 |
| Walk fast | 4061 | 4285 | 1.2920 | 0.7421 | 0.7931 |
| Walk and turn | 2136 | 2870 | 1.1528 | 0.7481 | 0.7560 |
| Marching | 10931 | 11710 | 1.3989 | 0.9258 | 0.9220 |
| Average | – | – | 1.1408 | 0.7764 | 0.8041 |

Table 8: Comparison of real-world and simulation performance

We observe that the real-world tracking error is slightly larger, likely due to starting from a random initial pose in reality and the inherent sim-to-real gap. Overall, the discrepancy between simulation and real is only 0.0117 rad per joint, which is acceptable and demonstrates robust simulation-to-real transfer.

## H   TRACKING PERFORMANCE COMPARISON

To clarify, BFM-Zero is, to our knowledge, the first to apply unsupervised RL to humanoid control capable of performing diverse tasks zero-shot. Prior frameworks typically rely on task-specific tracking rewards and focus purely on a single task: general motion tracking. In contrast, BFM-Zero is a fundamentally different method that constructs a unified latent space that captures reusable skills and supports reward optimization and discontinuous goal reaching. Nevertheless, we compared our method with the SOTA general motion tracking controller GMT Chen et al. (2025) in tracking performance. We use their released checkpoint and test on the test set in the same environment in MuJoCo. Using their released checkpoint, we evaluate $E_{mpjpe}$ on the test set in the same MuJoCo environment. As GMT Chen et al. (2025) cannot naturally stand up, the metric is computed only over motion segments where the episode has not yet been terminated.

| Methods | Tracking Error ($E_{\mathrm{mpjpe}}$) ($\downarrow$) | | Number of Samples ($\downarrow$) |
|---|---|---|---|
| | LAFAN1 | AMASS | |
| GMT | 2.2425 | 1.9064 | 6800M |
| **BFM-Zero** | **1.0789** | **1.0342** | 200M |

Table 9: Comparison of general tracking methods and **BFM-Zero**

In particular, we observe that GMT struggles with get-up motions and other on the-ground actions, significantly impairing its tracking performance on the test set. Even in terms of tracking accuracy,

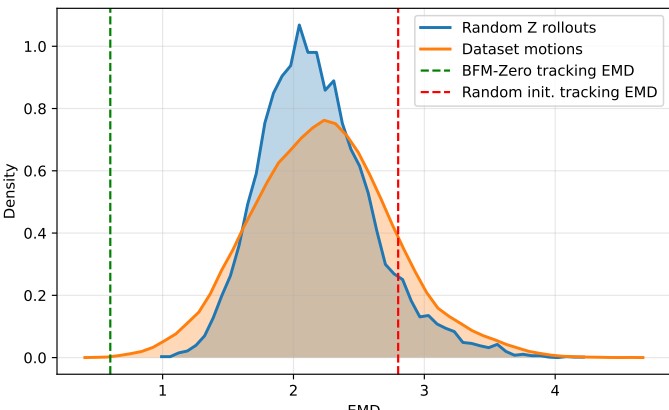

Figure 19: Distribution of EMD distances between two rollouts from two randomly sampled latents.

our results are comparable to GMT Chen et al. (2025) (Table 9), which already outperforms Exbody2 Cheng et al. (2024) and OmniH2O He et al. (2025b) in their respective evaluations. More importantly, BFM-Zero clearly surpasses GMT in fully out-of-distribution scenarios—such as external pushes, falls, or random initializations—where GMT typically fails to recover. In contrast, BFM-Zero naturally regains stability and continues to track robustly, due to its well-regularized, smooth, and dynamically aware skill space with even fewer samples.

## I  LATENT SPACE INVESTIGATION

To study the latent space more quantitatively, we replicate latent space studies done in the original FB-CPR work (Tirinzoni et al., 2025). Overall, these results indicate that, much like FB-CPR, BFM-Zero latent space contains an equally diverse set of motions as the training set, while covering the motions of the training set but also capturing motions not included in the training set. The latent space is clustered semantically by motion type, with clearly separated clusters.

**Behavior diversity.**   We sample $10,000$ pairs of random latents (z's), roll them out for 500 steps (10 seconds), and calculate the EMD of the two rollouts. For perspective, we also plot the distribution of EMDs between all the training motions, and include tracking evaluation performance as EMD on LAFAN1 dataset, both for randomly initialized policy and final version of the model. If the latent space maps only to a small set of motions, we expect to see small EMDs. As we see in Figure 19, the distributions of random rollouts and motions overlap with similar shape, indicating that BFM-Zero latent space captures the diversity of motions of the training set. Some of the random motions are further apart than randomly initialized policy from training motions (EMD $> 2.8$, $8\%$), further indicating two randomly sampled latents can have very distinct motions.

**Coverage of the training dataset**   We sample random 1000 random z's, roll them out for 500 steps, and calculate EMD to all the motions in the training LAFAN1 dataset (we use the dataset split into 10s motions). We then label each motion in the dataset "covered" if it has the smallest EMD to at least one random rollout. We then loosen this criteria by adding progressively increasing margin $m$ around each motion: if motion's distance to rollout is at most $m$ larger than the distance to the closest motion, then motion is counted as "covered". Ideally, randomly sampled motions should cover $100\%$ of datasets in this way with a small margin. In Figure 20, we see that 1000 random rollouts cover $26\%$ of the 862 motions in this way at zero margin, and by $0.7$ margin all motions are covered. EMD of $0.7$ is slightly higher than final tracking performance EMD ($0.6$) of the model, where the model is able to replicate most motions accurately. This result indicates that the model covers the dataset of motions. We also plot the distribution of the EMD to the closest motion. Even if the random samples cover all motions, they may be only replicating those specific motions (i.e., the latent space only covers the training motions). In Figure 20, we see that the EMD distance spans

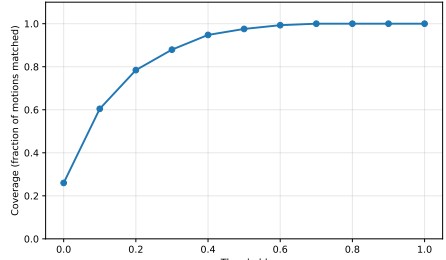 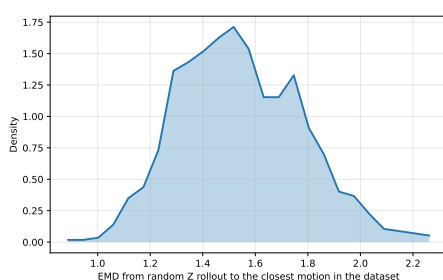

Figure 20: Coverage of the dataset by randomly sampling behavior latents, rolling them out and matching the trajectories with the closest motion (left), and the distribution of distances to the closest motions (right).

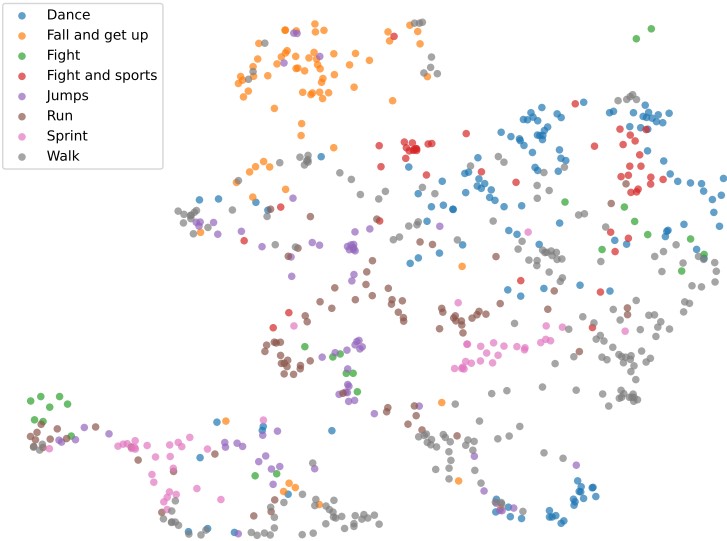

Figure 21: UMAP dimensionality reduction of LAFAN1 motions embedded into BFM-Zero's latent space.

$1.0 - 2.2$, with mode at $1.5$, which indicates that most randomly motions are significantly different from their closest match in the dataset, while some ($EMD > 2$) being almost as far as the average distance of two randomly sampled motions.

**Latent space clustering.** To study the clustering of motions in the latent space, we embed the motions in the LAFAN1 dataset using $ER_{FB}$ (Tirinzoni et al., 2025) (embed every frame of the motion and then average over). In FB-CPR work, authors took only middle part of the motions, but here we embed the whole motions as they are already separated into 10-second clips of the full motions. This will lead some points to be clustered close together (from the same motion), but also lead to potential overlap (e.g., jumping and walking motions share overlapping sequences of person standing still). Figure 21 shows the clustering of the points, with different regions for "fall and get up" and "dance". For quantitative numbers, we create a centroid embedding for each motion class by averaging the embeddings of that motion, and then assign new labels for each embedding based on which is the closest centroid. This mapping reaches $78.8\%$ accuracy and $0.598$ normalized mutual information (using `normalized_mutual_info_score` function from scikit-learn package (Pedregosa et al., 2011)), which indicates of clustered latent space by motion groups.

