# OpenReview forum: "BFM-Zero: A Promptable Behavioral Foundation Model for Humanoid Control Using Unsupervised Reinforcement Learning"
_ICLR.cc/2026/Conference — ICLR 2026 Poster_

### Official Review · Reviewer_qyhf · 2025-10-27

**Soundness:** 3
**Presentation:** 3
**Contribution:** 3
**Rating:** 6
**Confidence:** 2

**Summary:**

The authors propose an approach for Unsupervised RL pretraining of Behavior Foundational Models (BFMs) using a recently proposed technique called Forward-Backwards (FB) representation. The approach presented by the authors can essentially be framed as putting together existing techniques and scaling FB method to humanoid whole-body control, so that it could be prompted for different tasks without retraining. Moreover, they even propose a technique for fast adaptation, finetuning and high-level planning of their model.

**Strengths:**

The authors successfully put together a method from existing related approaches, such that it scales well to complex humanoid whole-body control. Their approach can successfully be used in zero-shot manner, as well as being used for fast adaptation and finetuning. This is a good empirical contribution since existing methods do not enjoy such properties.

**Weaknesses:**

I think authors could do a better job for a reader to make them understand how this work differs from existing approaches such as [1] and what are the most important differences which make their method work. Currently, the proposed approach differs in a few components:
* Different RL algorithm
* Section 2.2 also proposes a few techniques such as Domain Randomization (DR) and Reward Regularization (RR)

I think disentangling these design choices is important for a reader to understand why this approach works better. I think having additional ablations on DR and RR will greatly improve paper's technical presentation.


[1] Zero-Shot Whole-Body Humanoid Control via Behavioral Foundation Models, Andrea Tirinzoni, Ahmed Touati, Jesse Farebrother, Mateusz Guzek, Anssi Kanervisto, Yingchen Xu, Alessandro Lazaric, Matteo Pirotta, 2025

**Questions:**

What are design choices in this approach which make it work significantly better than previous approaches?

---

> ### Author Response · Authors · 2025-11-22
> **Author Response**
>
> We appreciate the reviewer’s thoughtful feedback, and we are glad that the reviewer finds our approach and empirical contributions meaningful.
>
> ## Ablations on DR and RR.
> > W1: I think disentangling these design choices is important for a reader to understand why this approach works better. I think having additional ablations on DR and RR will greatly improve the paper's technical presentation.
>
> We thank the reviewer for the suggestion. While our work builds on the FB-CPR [1] framework, the key contribution lies in exploring the extension of unsupervised RL to real robots, enabling robots to perform diverse tasks zero-shot **under a new training paradigm**. The Domain Randomization (DR) and Reward Regularization (RR) terms are primarily guided by the robot’s physical limits and the sim-to-real gap. For example, we penalize large actions because the motors become highly nonlinear near joint limits, whereas the simulator does not fully capture this behavior. In other words, these terms are **largely driven by physics, rather than our manual design**. Also, we have relatively simple DR to eliminate the foundational sim-to-real discrepancies.
>
> Nevertheless, we added results for models trained with and without auxiliary rewards in the revised version of the paper. By inspecting the training curves related to auxiliary penalties, we can clearly see that these metrics (e.g., action rate and feet orientation) are optimized *(Figure 16)* by the model using auxiliary rewards.
>
> In contrast, evaluation metrics in simulation *(see Figure 15)* remain very similar regardless of whether auxiliary rewards are included. In some cases, performance in simulation may even appear to improve when auxiliary rewards are removed (see table in *Figure 15* or below). However, this improvement does not translate to the real robot.
>
> | Auxiliary Reward | Tracking ($\downarrow$) | Goal ($\downarrow$) | Reward (LAFAN1) ($\uparrow$) |
> |:----------------:|:-----------------------:|:--------------------:|:-----------------------------:|
> | Yes              | 0.9806                  | 0.9673               | 273.1                         |
> | No               | 0.9572                  | 0.9664               | 287.9                         |
>
>
> In the real-world experiments, we have tested ablating the auxiliary rewards and domain randomization individually. Although the performance in simulation remains similar, the policies **typically fail immediately on real hardware**: (1) Without DR, the robot frequently exhibits foot slipping. (2) Without auxiliary rewards, the policy produces severe jittering on the real robot.
>
> ##  Design Choices.
>
> > Q1: What are design choices in this approach which make it work significantly better than previous approaches?
>
> This work overcomes the limitations of previous work [1], which was solely tested in simulation. Moving from a fully simulated environment to the real-world deployment is **far from being trivial and requires explicit adjustments and careful design** to the approach. Techniques such as DR and RR are fundamental for the success of the deployment. As shown in the ablations on RR, research iterations become challenging since the performance in simulation is not always a good indication of the real-world performance. For example, metrics slightly improve when training without RR. However, the robot almost immediately fails in the real robot, e.g., due to foot misplacement. The role of RR and DR is indeed to avoid overfitting on the simulator.
>
> Concerning other design choices, we added further ablations in the appendix related to model size, data composition and z dimension *(Appendix F)*. The key messages are: 1) residual networks are better than standard MLP networks; 2) the more diverse the motion dataset is, the better the downstream performance; 3) reward inference performance is highly influenced by the data distribution (inference of motions is, on average, better than using the replay buffer).
>
> [1] Zero-Shot Whole-Body Humanoid Control via Behavioral Foundation Models, Andrea Tirinzoni, Ahmed Touati, Jesse Farebrother, Mateusz Guzek, Anssi Kanervisto, Yingchen Xu, Alessandro Lazaric, Matteo Pirotta, 2025

---

> > ### Comment · Reviewer_qyhf · 2025-11-24
> > **Response**
> >
> > I would like to thank the authors for their response and for clarifying my.
> >
> > Thank you for explaining the role of DR and RR. Indeed, making the method work on real world robot is a non-trivial task. I'm increasing my score.

---

### Official Review · Reviewer_NnXb · 2025-10-29

**Soundness:** 3
**Presentation:** 2
**Contribution:** 2
**Rating:** 6
**Confidence:** 4

**Summary:**

This work presents BFM-Zero, an unsupervised RL method that learns a well-structured latent representation of whole-body motions. The authors hereby demonstrate zero-shot task performances on a physical humanoid, bringing unsupervised RL and forward-backward models to actual hardware. BFM-Zero is trained using a motion capture dataset to regularize the large space of behaviors towards more useful behaviors, reducing the search space immensely. It then allows for solving various tasks such as motion tracking or goal reaching in a zero-shot fashion, as well as allows for few-shot adaptation in the real world, where the environment can be modified, for example, by adding additional weight onto the robot. The paper shows that adding common tricks in sim-to-real transfer, such as domain randomization or privileged information into FB-CPR, can successfully transfer to the real world.

**Main contributions:**
- Extending the FB-CPR work to real robots, which includes adding Domain Randomization (DR) and an auxiliary rewards
- Presenting a few-shot adaptation via sampling-based optimization for hard-to-achieve tasks.

**Explanation of rating:** Overall, the paper is a first of its kind and shows zero-shot RL on a humanoid robot for whole-body control. However, from a technical perspective, it is not fundamentally different from previous work, such as MetaMotivo (Tirinzoni et al., 2024), and applies previously known techniques for sim-to-real transfer. While its main contribution lies in the transfer to the physical world, it lacks quality in evaluating results there. Given that this is one of the first attempts to bring unsupervised RL and FB to real humanoids, I still lean towards acceptance, but advise the authors to significantly improve the quality of the exposition and extend the real-world evaluation to provide a more substantial contribution.

**Strengths:**

- The authors show promising results for Zero-shot RL on humanoids. The video presents target pose tracking and continuous dancing, as well as few-shot adaptation in the real world on a challenging hopping example.
- The results are interesting. Especially the natural recovery, the few-shot adaptation, and motions with whole-body contact are impressive.

**Weaknesses:**

In general, the paper seems rushed and contains many sections with obvious errors (see below for minor issues caught during the read). This is also true for the evaluation, which raises more questions than it answers (see the question section).

- **Major:** The main contribution of this work is the presentation of a method that works on a real humanoid. However, beyond the few-shot adaptation task, there is little quantitative evaluation in the real world. Furthermore, the results presented on the robot do not demonstrate the method's limits. While it is clear that this formulation comes with many new advantages, I would be interested in understanding where the limits and gaps are from a motion tracking perspective. We have seen quite expressive motion-tracking capabilities in, e.g., He et al., ASAP; how close do we get to such performances? How much expressivity is encoded? Different experiments could help understand the performance in the real world as well as help understand/interpret the gap between the results presented in Fig. 3 and real-world results, e.g., how much can we trust those numbers for the real world? For example, to better understand the sim-to-real gap, it would be helpful to pick 1-2 examples and compute the MPJPE in the real world. This would give a sense of how much the metrics might vary between simulation and reality.
- A Related Work: In general, the related work is incomplete, rushed, and oversimplifies the progress in the field of robotics. There has been significant work before 2025 that has investigated and enabled versatile human motion on robots.
    - DR is a well-explored technique with methods dating back to 2017, such as (Peng et al., Sim-to-Real Transfer of Robotic Control with Dynamics Randomization)
    - Bringing bigger datasets of whole-body human motion beyond locomotion to robots has been investigated before 2025 in many works, some also identified by the authors (He et al., 2024, HOVER: Versatile Neural Whole-Body Controller for Humanoid Robots), but also some not mentioned in this section, such as (Cheng et al., 2024, Expressive Whole-Body Control for Humanoid Robots), (He et al., 2024, OmniH2O: Universal and Dexterous Human-to-Humanoid Whole-Body Teleoperation and Learning) or not included at all, such as (Serifi et al., 2024, VMP: Versatile Motion Priors for Robustly Tracking Motion on Physical Characters), (Fu et al., 2024, HumanPlus Humanoid Shadowing and Imitation from Humans), (Dugar et al., 2024, Learning Multi-Modal Whole-Body Control for Real-World Humanoid Robots), (He et al., 2024, Learning Human-to-Humanoid Real-Time Whole-Body Teleoperation), most of this work is from robotics conferences.

- Minor:
    - L139, L996: broken reference.
    - Figure 3: missing values.
    - Some tables are figures.

**Questions:**

- In the videos, the robot seems a bit unstable, often drifting in a direction. The authors report MPJPE metrics; however, this does not include the root linear and angular velocities, which seem quite off, judging from the videos. It is not clear to me why the root linear velocity is not part of the observable state. I would assume this could mitigate some of this behavior. Could the authors elaborate on this design choice?
- The Auxiliary critic is a new component added to the framework. However, it is not ablated in any of the results. How important is this part for the sim-to-real transfer at the end? In general, how important is DR? While reading, I was wondering how FB-CPR would perform on a robot, and which components in this work represent the main changes?
- The distribution of the rewards seems very interesting (Fig. 3), especially the observation that there seem to be severe collapses. The authors hypothesize that domain randomization has led to this. move-ego-0-0 appears to be a static pose (judging from the supplementary video where the motion is shown). If the hypothesis of the authors is true, why do we only see this in some of the motions? Investigating for which behaviors this occurs would be interesting.
- The evaluation of the latent space is minimal. e.g., the interpolation result is not surprising; most likely, all those states in between for raising the hand are part of the dataset as well. What happens if we interpolate between dancing and crouching motions? How does the robot's behavior change in those more challenging scenarios where we interpolate trajectories in space and time?
- The results on the Booster T1 robot are incomplete. It would be interesting to understand if there are weaknesses of the method when applied to this robot, beyond the obvious mechanical limitations. For example, do the robot's specific capabilities affect the training of BFM-Zero in any way (e.g., do the discriminators remain similarly stable)?
- It would also be helpful to see the selected AMASS goal poses for the goal-based evaluation (Fig. 13).
- It is not clear to me why the authors selected 175 motions from the CMU subset of AMASS. This subset contains many walking cycles, which, according to the latent-space plots, appear to be well covered. In simulation, it would be straightforward to test on the entire AMASS dataset, thereby improving understanding of out-of-distribution behavior.
- Why did the authors not train on the much larger AMASS data? How do the discriminators scale with more data, or do we observe mode-collapse?

---

> ### Author Response · Authors · 2025-11-22
> **Author Response (Part1/3)**
>
> We sincerely thank the reviewer for the detailed feedback and constructive suggestions. We appreciate the recognition of the zero-shot performance and real-world demonstrations. Below we address the main concerns.
>
> ## About real world validation.
>
> We thank the reviewer for the suggestion. Previously, we demonstrated the real-world performance of BFM-Zero through **a robust, uninterrupted video take** (In the external link, we further include a single-take demos containing tracking, reward, goals, from the same policy), showing zero-shot reward optimization, goal transitions, robust tracking, emergent recovery behaviors, and payload handling. This clearly illustrates that BFM-Zero functions as a generalist capable of diverse tasks on real robots, providing strong sim-to-real evidence.
>
> We further report E_MPJPE on the real robot (examples shown in our video), which is slightly larger than simulation results, likely due to starting from a random initial pose in reality and the inherent sim-to-real gap. Overall, the discrepancy between simulation and real is only 0.0117 rad per joint, which is acceptable and demonstrates **robust simulation-to-real transfer**.
>
> | Motions        | Start Frame | End Frame | $E^{\text{real}}_{\text{mpjpe}}$ ($\downarrow$) | $E^{\texttt{sim-nodr}}_{\text{mpjpe}}$ ($\downarrow$) | $E^{\texttt{sim-dr}}_{\text{mpjpe}}$ ($\downarrow$) |
> |:--------------:|:-----------:|:---------:|:-----------------------------------------------:|:----------------------------------------------------:|:---------------------------------------------------:|
> | Dance-1        | 1           | 2400      | 1.1945                                          | 0.7631                                               | 0.7736                                              |
> | Dance-2        | 1           | 2000      | 1.0003                                          | 0.9047                                               | 0.9235                                              |
> | Walk slowly    | 1           | 320       | 0.8662                                          | 0.5746                                               | 0.6561                                              |
> | Walk fast      | 4061        | 4285      | 1.2920                                          | 0.7421                                               | 0.7931                                              |
> | Walk and turn  | 2136        | 2870      | 1.1528                                          | 0.7481                                               | 0.7560                                              |
> | Marching       | 10931       | 11710     | 1.3989                                          | 0.9258                                               | 0.9220                                              |
> | **Average**    | --          | --        | 1.1408                                          | 0.7764                                               | 0.8041                                              |
>
>
> ## Limitation Analysis and Expressiveness.
>
> We thank the reviewer for the comment. We acknowledge that BFM-Zero may not achieve the same level of motion-tracking accuracy as task-specific methods such as ASAP[1], BeyondMiMic[2] or TWIST[3], which focus on highly expressive tracking and sim-to-real alignment for **a single motion**. However, BFM-Zero is trained as a generalist for diverse zero-shot tasks, and the use of unsupervised learning enables new capabilities such as zero-shot reward inference, smooth and safe transitions, natural recovery, skill composition and robust adaptation. With this general motion space, it is possible to do **further fine-tuning or efficient optimization to achieve higher precision** or more challenging motions, as we have demonstrated in the few-shot adaptation sessions. At the same time, we acknowledge the current limitations of our framework: it is still a blind policy without object interaction, and reward inference can occasionally be unstable in rare states. Extending this approach to broader settings remains an open research problem.
>
> ## Related works and minor errors.
> We thank the reviewer for the detailed feedback on related work and minor issues. We agree that the Related Work section can be expanded to more comprehensively reflect prior contributions in robotics, and we include additional references in the revision (Appendix.A).
>
> Regarding minor issues, we will correct broken references, properly format tables and figures. We will clarify Figure 3: the single numbers shown represent “reward inference” results (not differences computed on AMASS or LaFan datasets) rather than missing values, and we will explain this in the revised version.

---

> ### Author Response · Authors · 2025-11-22
> **Author Response (Part 2/3)**
>
> ## Questions
> > Q1: In the videos, the robot seems a bit unstable, often drifting in a direction. ... It is not clear to me why the root linear velocity is not part of the observable state. ... Could the authors elaborate on this design choice?
>
> We acknowledge that the real-world performance shows some global drift, but this is expected since our task focuses on local tracking rather than using full odometry for global pose recovery. In practice, root velocity is also difficult to obtain accurately in real-world settings. As shown in Figure 3 in the paper, we additionally trained a variant with root velocity as input (called BFM-Zero-priv), and its performance remains comparable.
>
> > Q2: The Auxiliary critic is a new component added to the framework. How important is this part for the sim-to-real transfer at the end? In general, how important is DR? ...
>
>
> We thank the reviewer for the suggestion. These terms are primarily guided by the robot’s physical limits and the sim-to-real gap. For example, we penalize large actions because the motors become highly nonlinear near joint limits, whereas the simulator does not fully capture this behavior. In other words, these terms are largely driven by physics, rather than our manual design. Also, we have relatively simple DR to eliminate the foundational sim-to-real discrepancies.
>
> Nevertheless, we added results for models trained with and without auxiliary rewards in the revised version of the paper. By inspecting the training curves related to auxiliary penalties, we can clearly see that these metrics (e.g., action rate and feet orientation) are optimized *(Figure 16)* by the model using auxiliary rewards.
>
> In contrast, evaluation metrics in simulation *(see Figure 15)* remain very similar regardless of whether auxiliary rewards are included. In some cases, performance in simulation may even appear to improve when auxiliary rewards are removed (see table in *Figure 15* or below). However, this improvement does not translate to the real robot.
>
> | Auxiliary Reward | Tracking ($\downarrow$) | Goal ($\downarrow$) | Reward (LAFAN1) ($\uparrow$) |
> |:----------------:|:-----------------------:|:--------------------:|:-----------------------------:|
> | Yes              | 0.9806                  | 0.9673               | 273.1                         |
> | No               | 0.9572                  | 0.9664               | 287.9                         |
>
>
> In the real-world experiments, we have tested ablating the auxiliary rewards and domain randomization individually. Although the performance in simulation remains similar, the policies **typically fail immediately on real hardware**: (1) Without DR, the robot frequently exhibits foot slipping. (2) Without auxiliary rewards, the policy produces severe jittering on the real robot.
>
> > Q3: The distribution of the rewards seems very interesting (Fig. 3), especially the observation that there seem to be severe collapses. ...Investigating for which behaviors this occurs would be interesting.
>
> We have added a more in-depth investigation of this issue in the appendix. Reward inference can be performed using any state-based dataset; it is only necessary to relabel the entries with the desired reward function. This means that we are not limited to using the replay buffer built by the agent, but can also directly use the motions. Our quantitative evaluation (*Figure 13, 14*) shows that performance with respect to reward inference increases (on average) when using motions rather than the replay buffer. Inference with LAFAN1 leads to better policies and thus returns on tasks covered in the dataset (e.g., moving forward, backward, on the side). However, we are able to infer better policies for “non-standard” tasks (e.g., rotate, crouching, move with low height) when using the experience self-generated by the agent because these samples are collected by directly deploying policies learned by the agent. Selecting the optimal dataset for reward inference could be an interesting direction for future research.

---

> ### Author Response · Authors · 2025-11-22
> **Author Response (Part 3/3)**
>
> >Q4: The evaluation of the latent space is minimal.
>
> We thank the reviewer for the feedback. While the hand-raising interpolation may include intermediate states in the dataset, the fact that simple SLERP interpolation produces a semantically smooth and coherent transition still demonstrates that our latent space is highly continuous and interpretable. To further address the concern, we conducted additional dynamic interpolations (video on the webpage link). For example, interpolating between latents for moving left 0.3 m/s and moving right 0.3 m/s produces a smooth progression: leftward velocity decreases, approaches near zero at the midpoint, and then increases rightward. Another example is dancing combined with backward crouching. With a small fraction of crouch, the base height is slightly lowered, and the robot leans back like in a crouch while maintaining dancing motions in the upper body. As the proportion of crouch increases, the robot fully crouches, while the main body exhibits rhythmic motion. This shows that even for dynamic motions, our latent space yields meaningful and stable spatial–temporal interpolations.
>
>
>
> Furthermore, we add lots of space analysis in *Appendix I*. these results indicate that, much like FB-CPR, BFM-Zero latent space contains an equally diverse set of motions as the training set (first result), while covering the motions of the training set but also capturing motions not included in the training set (second result). The latent space is clustered semantically by motion type, showcased both by dimensionality reduction and quantitative numbers (third result).
>
> >Q5. The results on the Booster T1 robot are incomplete.
>
> We thank the reviewer for the question. BFM-Zero shows **stable discriminator training and satisfactory convergence** on T1(See Figure 11.b), and we include the corresponding training curves of T1 as evidence. We are also trying to do real-world T1 experiments that will be included in camera-ready upon acceptance.
>
> > Q6-8. Related to AMASS.
>
> We thank the reviewer for the insightful questions. We will include the selected AMASS goal poses in the appendix for clarity (Figure 10). The entire AMASS dataset is out-of-distribution relative to our LaFAN-trained model. The 175 CMU motions were simply randomly selected for convenience. To address the concern, we also ran experiments with larger AMASS subsets and AMASS + LaFAN jointly. Evaluations on AMASS (remove the training dataset) show stable latent-space behavior, without discriminator instability or mode collapse (Figure 12, 13). These results and clarifications will be added in the revision.
>
>
> >[1] ASAP: Aligning Simulation and Real-World Physics for Learning Agile Humanoid Whole-Body Skills, Tairan He, Jiawei Gao, Wenli Xiao, Yuanhang Zhang, Zi Wang, Jiashun Wang, Zhengyi Luo, Guanqi He, Nikhil Sobanbab, Chaoyi Pan, Zeji Yi, Guannan Qu, Kris Kitani, Jessica Hodgins, Linxi "Jim" Fan, Yuke Zhu, Changliu Liu, Guanya Shi, 2025
> [2] BeyondMimic: From Motion Tracking to Versatile Humanoid Control via Guided Diffusion, Qiayuan Liao, Takara E. Truong, Xiaoyu Huang, Yuman Gao, Guy Tevet, Koushil Sreenath, C. Karen Liu, 2025
> [3] TWIST: Teleoperated Whole-Body Imitation System, Yanjie Ze, Zixuan Chen, João Pedro Araújo, Zi-ang Cao, Xue Bin Peng, Jiajun Wu, C. Karen Liu

---

> ### Comment · Reviewer_NnXb · 2025-11-27
> **Response**
>
> I thank the authors for addressing the questions comprehensively. The revisions have significantly strengthened the evaluation, providing much more insight and answering key open questions. I am now fully satisfied and believe this work will serve as an excellent example for future research in BFMs. I have increased my score accordingly.

---

### Official Review · Reviewer_qsJd · 2025-11-01

**Soundness:** 2
**Presentation:** 2
**Contribution:** 2
**Rating:** 4
**Confidence:** 4

**Summary:**

The paper trains a single latent-conditioned policy built on a forward-backward (FB-CPR) backbone, that claims to handle motion tracking, sparse-reward tasks, and goal-pose reaching with no task-specific fine-tuning. Training is fully off-policy in Isaac Sim with heavy domain randomisation. The same network is then deployed zero-shot on a Unitree G1, with optional latent-space CEM/dual-loop annealing (Dial-MPC) for quick adaptation. Evaluations span simulations and real hardware, highlighting robustness in balance, recovery, and payload handling.

**Strengths:**

## Strengths

- Real World Demonstrations: Impressive zero-shot performance on the Unitree G1, including balance maintenance, push recovery from large perturbations, and handling a 4 kg payload, showcases practical sim-to-real transfer.
- Clear Description of Method: The FB backbone, critics, reward shaping, and auxiliary components are explained adequately, though familiarity with prior FB-CPR work is helpful for full understanding.
- Thorough Ablations: Quantitative evaluations of privileged vs. proprioceptive sensing, domain randomization effects, and sim-to-sim robustness (e.g., Isaac to MuJoCo) provide strong evidence for the design choices.

**Weaknesses:**

## Weaknesses

- Lack of Competitive Baselines: The paper does not compare against state-of-the-art methods like Ex-Body 2, OmniH2O, or Puppeteer on identical tasks and metrics, making it hard to gauge relative advancements.

- Under-Quantified Latent Space Claims: T-sne plot of latents looks interpretable, But assertions about the "promptable" and semantic nature of the latent space are not backed by quantitative measures, weakening the foundation model claims.

**Questions:**

## Questions / Requested Revisions

- Please add comparisons to competitive baselines (e.g., re-run Ex-Body 2, OmniH2O, Puppeteer on the same tasks and report identical metrics like MPJPE for tracking).
- Expand the evaluation to include error tracking, and behaviour diversity metrics. Provide full breakdowns in the appendix.
- Quantify the latent space semantics with metrics such as cluster-purity scores, linear-probe accuracy for task prediction, or mutual information to substantiate claims.
-  Release training artifacts, including scripts, Docker images, MPC optimization parameters, and safety limits used on the G1, to facilitate reproducibility.


Overall, the work extends BFM to real humanoids with solid engineering, but substantial overlap with prior BFM reduces novelty.
Stronger sim-to-real evidence via detailed real-robot metrics, baselines, and stress tests is needed to convincingly demonstrate progress.

---

> ### Author Response · Authors · 2025-11-21
> **Author Response (Part 1/2)**
>
> We sincerely thank the reviewer for the detailed feedback and constructive suggestions. Below, we address the concerns in detail.
>
> ## W1&Q1: Compare to other baselines
>
> We thank the reviewer for the suggestion. To clarify, BFM-Zero is, to our knowledge, the first to apply unsupervised RL to humanoid control **capable of performing diverse tasks zero-shot**. Prior frameworks typically rely on task-specific tracking rewards and focus purely on a single task: general motion tracking. In contrast, BFM-Zero is a **fundamentally different** method that constructs a unified latent space that captures reusable skills and supports **reward optimization and discontinuous goal reaching**. Nevertheless, we compared our method with the SOTA general motion tracking controller [1] in tracking performance:
>
> In particular, we observe that GMT struggles with get-up motions and other on-the-ground actions, significantly impairing its tracking performance on the test set. Even in terms of tracking accuracy, our results are comparable to GMT, which already outperforms Exbody2 and OmniH2O in their respective evaluations. More importantly, BFM-Zero clearly surpasses GMT in fully out-of-distribution scenarios—such as external pushes, falls, or random initializations—where GMT typically fails to recover. In contrast, BFM-Zero naturally regains stability and continues to track robustly, **due to its well-regularized, smooth, and dynamically aware skill space with even fewer samples**.
>
> | Methods   | | Tracking Error |($E_{\text{mpjpe}}$)   | | Number of Samples |
> |:---------:|:-:|:---------------------------------:|:-:|:-:|:----------------:|
> |           |   | LAFAN1                             | AMASS  | |                 |
> | GMT       |   | 2.2425                             | 1.9064 | | 6800M            |
> | BFM-Zero  |   | **1.0789**                         | **1.0342** |  | **200M**        |
>
> For other methods that are not designed for real robots like Puppeteer  (i.e., without motor or joint constraints or with full observation), direct comparison is difficult because the settings differ significantly and they **cannot be deployed on hardware**.
>
>
> ## W2&Q3. Under-quantified latent space claims
>
> We thank the reviewer for the suggestion. We have added a section “Latent space investigation” in the appendix to address the comments. We run the experiments reviewer suggested, and replicate latent space studies done in the original FB-CPR work [2, Appendix I]. Overall, these results indicate that, much like FB-CPR, BFM-Zero latent space contains an equally diverse set of motions as the training set (first result), while covering the motions of the training set but also capturing motions not included in the training set (second result). The latent space is clustered semantically by motion type, showcased both by dimensionality reduction and quantitative numbers (third result).
>
> ## Q2: Expand of evaluations
> We thank the reviewer for the suggestion. First, we already include error tracking in our evaluation (E_mpjpe), showing per-joint tracking error (the table in *Figure 3*). Second, we demonstrate behavioral diversity in multiple ways: (1) by evaluating on an OOD dataset (the table in *Figure 3*, *Figure 12*), which proves that a broad range of skills are covered in the latent space, and (2) qualitatively, by showing that different reward prompts can induce diverse modes of behavior (Fig.6 (d)), highlighting the multimodal nature. We will add additional breakdowns in the appendix for clarity.
>
> ## Q4: Reproducibility and release
> Thank you for the question. We would like to emphasize that all video results were captured in single, uninterrupted takes, ensuring their reliability. To facilitate full reproducibility, we plan to release all training and deployment artifacts, including code, scripts, Docker images, parameters, and checkpoints.
>
> >
> [1] GMT: General Motion Tracking for Humanoid Whole-Body Control, Zixuan  Chen, Mazeyu Ji, Xuxin Cheng, Xuanbin Peng, Xue Bin Peng, and Xiaolong Wang, 2025
> [2] Zero-Shot Whole-Body Humanoid Control via Behavioral Foundation Models, Andrea Tirinzoni *et al.*, 2025

---

> ### Author Response · Authors · 2025-11-21
> **Author Response (Part 2/2)**
>
> ## Response to the **Overall** Comment
> We thank the reviewer for the feedback. While our work builds on the FB-CPR framework, the key contribution lies in **exploring the extension of unsupervised RL to real robots, by constructing a unified, goal-centric latent skill space**. This enables robots to perform diverse tasks zero-shot under a new training paradigm. Our real-world demonstrations—including zero-shot reward optimization, goal transitions, robust tracking, recovery, and payload handling—provide strong sim-to-real evidence. We further report E_MPJPE on the real robot (examples shown in our video), which is slightly larger (see table below or Table 8 in revision), likely due to starting from a random initial pose in reality and the inherent sim-to-real gap. Overall, the discrepancy between simulation and real is only **0.0117 rad per joint**, which is acceptable and demonstrates robust simulation-to-real transfer.
> | Motions        | Start Frame | End Frame | $E^{\text{real}}_{\text{mpjpe}}$ ($\downarrow$) | $E^{\texttt{sim-nodr}}_{\text{mpjpe}}$ ($\downarrow$) | $E^{\texttt{sim-dr}}_{\text{mpjpe}}$ ($\downarrow$) |
> |:--------------:|:-----------:|:---------:|:-----------------------------------------------:|:----------------------------------------------------:|:---------------------------------------------------:|
> | Dance-1        | 1           | 2400      | 1.1945                                          | 0.7631                                               | 0.7736                                              |
> | Dance-2        | 1           | 2000      | 1.0003                                          | 0.9047                                               | 0.9235                                              |
> | Walk slowly    | 1           | 320       | 0.8662                                          | 0.5746                                               | 0.6561                                              |
> | Walk fast      | 4061        | 4285      | 1.2920                                          | 0.7421                                               | 0.7931                                              |
> | Walk and turn  | 2136        | 2870      | 1.1528                                          | 0.7481                                               | 0.7560                                              |
> | Marching       | 10931       | 11710     | 1.3989                                          | 0.9258                                               | 0.9220                                              |
> | **Average**    | --          | --        | 1.1408                                          | 0.7764                                               | 0.8041                                              |
>
> Together, these results highlight the robustness and effectiveness of our approach for real-world humanoid control.

---

### Official Review · Reviewer_yhgB · 2025-11-01

**Soundness:** 3
**Presentation:** 3
**Contribution:** 3
**Rating:** 6
**Confidence:** 2

**Summary:**

BFM-Zero introduces a framework to learn humanoid control from unsupervised RL. As opposed to most approaches that define motion tracking rewards over a dataset, BFM-Zero allows the model to explore in an unstructured way and pushes its action distribution toward human-like motions with a GAN-style discriminator trained on motion capture data. The authors demonstrate real world deployments of BFM-Zero.

**Strengths:**

- The method contrasts the dominant PPO over a motion dataset approach and demonstrates that it works. This may present a path forward to locomotion models trained with less reward engineering.
- Smooth and structured latent space is likely helpful for multimodal prompting or other downstream tasks.
- Pretraining seems scalable, subject to limits of the simulation model.
-Real world sim-to-real deployment of off policy-trained model.

**Weaknesses:**

- "Foundation model" itself may be overclaiming since the model is essentially just a low level controller without rich vision or touch sensing.
- Prompting occurs in a human-uninterpretable latent space rather than something like language.
- More detailed comparisons of computational cost and representation quality with typical on-policy methods would be more convincing to validate the usefulness of such an approach.
- Policy quality is still upper-bounded by simulation environment

**Questions:**

I'm curious to learn more about the latent space design decisions. How does the dimensionality affect the ease of prompting and performance of the model?

Does the model learn diverse behaviors or does it mode-collapse like on-policy algorithms?

---

> ### Author Response · Authors · 2025-11-21
> **Author Response (Part 1/2)**
>
> We thank the reviewer for the constructive feedback. Below we address the concerns in detail.
>
> ## Foundation model and promptable
>
> > W1: "Foundation model" itself may be overclaiming.
>
> We respectfully clarify that our use of the term “Foundation Model”(FM) follows its **established definition** in recent literature, rather than an overclaim of broad generality. The term Behavior(al) Foundation Model (BFM) was first introduced in [1], where it refers to a model pretrained on unsupervised data that supports zero-shot execution of multiple tasks without retraining. Subsequent work [2, 3]  further consolidates this definition. Following this usage, many recent models—such as HOVER [4], MaskedMimic [5], and BFM4Humanoid [3]—are also categorized as BFMs[2] even though they lack full multimodal sensing, such as rich vision or touch. Thus, the term refers to a **behavior-level motor foundation model**—a generalizable controller that spans diverse skills and tasks—rather than a multimodal foundation model involving perception inputs.
>
> Our model follows this established usage: it is a pretrained model that handles diverse tasks and goal representations without retraining and can be conditioned for downstream control. This skill level BFM is the first step towards multi-modal humanoid FMs. We agree that the Foundation Model could be misleading to the general audience, so we will clarify this in the manuscript to avoid potential misunderstanding.
>
>
> > W2: Prompting occurs in a human-uninterpretable latent space rather than something like language.
>
> By “promptable,” we mean the ability to elicit behaviors encoded in the model using different modalities that **capture human intent at the control level**. While natural language is convenient for expressing high-level goals, reliably translating it into precise motor behaviors is challenging, as it often requires large paired datasets and can be ambiguous or insufficiently actionable. Accordingly, our work focuses on prompts that are natively usable without additional supervision. Our goal format, like reward functions, serves as **a fundamental interface between user intent and motor skills**. They are explainable, task-centric, and can be seamlessly integrated into an unsupervised RL pipeline. Importantly, existing language-to-motion or language-to-reward models [6, 7, 8] can be incorporated upstream, making it straightforward to connect BFM-Zero with other types of user interfaces, including natural language, via a simple prompting pipeline.
>
> ## Comparison with on-policy methods
>
> > W3: More detailed comparisons of computational cost and representation quality with typical on-policy methods.
>
> We thank the reviewer for the suggestion. To clarify, BFM-Zero is a **fundamentally different and sample efficient** training paradigm compared to prior on-policy frameworks that typically rely on PPO with task-specific rewards and focus purely on **a single task**, such as motion tracking. BFM-Zero constructs a large, reusable, smooth, and dynamic-aware latent skill space that supports zero-shot inference, natural transitions, and recovery, which are hard to achieve with prior methods.
>
> Nevertheless, to provide a comparison, we include tracking performance against SOTA on-policy methods like GMT [9] (Table 9 in revision). Notably, our results surpass GMT in **tracking accuracy and robustness**, enabled by a well-regularized, smooth, dynamically aware skill space with even **fewer samples**.
>
> | Methods   | | Tracking Error |($E_{\text{mpjpe}}$)   | | Number of Samples |
> |:---------:|:-:|:---------------------------------:|:-:|:-:|:----------------:|
> |           |   | LAFAN1                             | AMASS  | |                 |
> | GMT       |   | 2.2425                             | 1.9064 | | 6800M            |
> | BFM-Zero  |   | **1.0789**                         | **1.0342** |  | **200M**        |
>
> >[1] Zero-Shot Whole-Body Humanoid Control via Behavioral Foundation Models, Andrea Tirinzoni *et al.*, 2025
> [2] A Survey of Behavior Foundation Model: Next-Generation Whole-Body Control System of Humanoid Robots, Mingqi Yuan *et al.*, 2025
> [3] Behavior Foundation Model for Humanoid Robots, Weishuai Zeng *et al.*, 2025
> [4] HOVER: Versatile Neural Whole-Body Controller for Humanoid Robots, Tairan He *et al.*, 2024
> [5] MaskedMimic: Unified Physics-Based Character Control Through Masked Motion Inpainting, Chen Tessler *et al.*, 2024
> [6] MotionGPT: Human Motion as a Foreign Language, Biao Jiang *et al.*
> [7] GROVE: A Generalized Reward for Learning Open-Vocabulary Physical Skill, Jieming Cui *et al.*, 2024
> [8] AnySkill: Learning Open-Vocabulary Physical Skills for Interactive Agents, Jieming Cui *et al.*, 2024
> [9] GMT: General Motion Tracking for Humanoid Whole-Body Control, Zixuan Chen *et al.*, 2025
> [10] Track Any Motions under Any Disturbances, Zhikai Zhang *et al.*, 2025

---

> ### Author Response · Authors · 2025-11-21
> **Author Response (Part 2/2)**
>
> ## Upper-bounded by simulation env.
>
> > W4: Policy quality is still upper-bounded by simulation environment
>
> We agree that performance is influenced by simulation fidelity, similar to almost all existing humanoid learning approaches [3, 4, 9, 10]. That said:
>
> - The modern physics simulator we use already provides high-quality dynamics that transfer well to the real world, as shown in our real-world demonstrations.
> - Our pretrained model provides a strong initialization that is particularly suitable for efficient real-world fine-tuning. For example, if we want the robot to run faster in the real world, we can start from a running latent in BFM-Zero and adapt it through limited, safe real-world interactions (similar to Sec 3.3) rather than learning everything from scratch.
>
> Together, these properties make our approach practical and effective for real-world humanoid deployment.
>
> ## Latent space design decisions.
>
> > Q1: I'm curious to learn more about the latent space design decisions. How does the dimensionality affect the ease of prompting and performance of the model?
>
> To answer this question, we have run experiments with z dim equal to [32,64,128, 256, 384, 512] on 2 seeds. We kept all the other training parameters fixed and reported performance across tracking, reward optimization and goal reaching (Table below or *Table 7* in the revision). For low dimensions (i.e., 32, 64, 128) the performance degrades w.r.t. the nominal paper configuration (i.e., z dim = 256). This performance loss is observed in all the inference modalities (tracking, goal-reaching and reward). For larger dimensions (i.e., 384, 512) we don’t observe any performance degradation showing that the training is stable. Performance even increases a bit for tracking and reward. Additional ablations on other design choices are provided in *Appendix F*.
>
> | Z dimension | Tracking ($\downarrow$) | Goal ($\downarrow$) | Reward (LaFAN) ($\uparrow$) |
> |:-----------:|:------------------------:|:-------------------:|:----------------------------:|
> | 32          | 1.3963                   | 1.6870              | 213.8224                     |
> | 64          | 1.2455                   | 1.3141              | 240.4211                     |
> | 128         | 1.0512                   | 1.0830              | 263.9501                     |
> | 256         | 0.9806                   | 0.9673              | 273.0826                     |
> | 384         | 0.9266                   | 0.9704              | 290.9922                     |
> | 512         | 0.9232                   | 0.9758              | 307.3982                     |
>
>
> ## Diversity or Mode-collapse
>
> > Q2: Does the model learn diverse behaviors, or does it mode-collapse like on-policy algorithms?
>
> **Our model learns diverse behaviors rather than collapsing into a narrow mode.** During unsupervised pretraining, the policy is explicitly encouraged to explore the entire behavior space through a random latent sampling strategy (when we perform rollouts, a certain proportion of latents are randomly sampled from the hypersphere). This results in (1) strong tracking performance on both the pretraining dataset and OOD motions (the table in Figure 3, Figure 12), indicating broad skill coverage, and (2) the ability for a single reward prompt to induce multiple valid behaviors (Figure 6 (d)), demonstrating that the learned latent space retains meaningful diversity instead of collapsing.

---

> > ### Comment · Reviewer_yhgB · 2025-11-27
> >
> > Thank you for the detailed response. I am raising my score.

---

### Author Response · Authors · 2025-12-02
**Summary of our rebuttal**

We thank the Area Chair for reviewing our paper again. This paper presents BFM-Zero, an unsupervised RL framework that constructs a structured and dynamics-aware latent space for whole-body humanoid motion. The learned latent enables zero-shot task performance on real hardware and few-shot adaptation under real-world variations. As the first-of-its-kind model, BFM-Zero marks a concrete step toward scalable, promptable behavioral foundation models for whole-body humanoid control.

All reviewers recognized the strengths of our work. They noted that our approach effectively advances humanoid whole-body control, demonstrates **impressive zero-shot and few-shot real-world performance**, enables natural recovery and robust motion execution, and provides **good empirical contributions not seen in existing methods** [R_yhgB, R2_NnXb, R_qsJd, R_qyhf]. The method’s smooth latent space [R_yhgB, R_qsJd], clear design, thorough ablations [R_qsJd] were also appreciated. Overall, reviewers acknowledged both the empirical contributions and the potential of our approach to support downstream tasks and practical sim-to-real transfer.

For clarity during the review, we summarize the main concerns raised by each reviewer and how we address them. Additional questions are addressed in detail within the discussion.
- Further explanation of the “promptable foundation model” (R_yhgB) → [See our explanation below]
- Additional comparison with baseline on-policy methods(R_yhgB, R_qsJd) → [Appendix H]
- Evaluation in real-world settings is suggested (R_NnXb, R_qsJd) → [Appendix G]
- Additional ablations on design choices, domain randomization, and auxiliary rewards(R_qyhf) → [Appendix F]
- Further analysis of the latent space (R_qsJd, R_NnXb) → [Appendix I]

We present the rebuttal timeline below:
- Around Nov 22 [EST], we submitted all our rebuttals.
- Nov 24, 12:04 [EST]: R_qyhf provided a **positive** response, acknowledging the non-trivial nature of our task and confirming that their concerns regarding our contribution were **addressed**. *Increase score 6 to 8*
- Nov 27, 04:00 [EST]: R_NnXb confirmed that the revisions had significantly **strengthened** the evaluation and expressed full satisfaction, noting that this work will **serve as an excellent example**. *Increase score 6 to 8*
- Nov 27, 04:05 [EST]: R_yhgB provided a **positive response**. *Increase score 6 to 8*
- Nov 27, 10:09 [EST]: The issue reported (according to OpenReview statement). R_qsJd is still reviewing our rebuttal and has not yet provided a response.

Before the reported issue, the scores were:
- R_qyhf: 8
- R_NnXb: 8
- R_yhgB: 8
- R_qsJd: 4

We thank the Area Chair again for the time and consideration. Given the reviewers’ positive follow-ups and the clarifications we have provided, we hope the paper is now viewed as a constructive and valuable contribution.

---

### Meta-Review · Area_Chair_u7PE · 2026-01-12

**Summary:**

The paper proposes BFM-Zero, a behavioral foundation model that utilizes unsupervised reinforcement learning—specifically Forward-Backward (FB) representations—for humanoid whole-body control. The framework embeds motions, goals, and rewards into a dynamics-aware latent space, allowing a single policy to be prompted for various downstream tasks without retraining. Reviewers generally lauded the impressive real-world demonstrations on a Unitree G1 robot, highlighting its capacity for balance maintenance, payload handling, and natural recovery from large external perturbations.

**Reviewer Concerns:**

Following major concerns were addressed by the rebuttal.
- Comparison with Baselines (Reviewer yhgB, qsJd): The authors provided a head-to-head comparison with GMT (General Motion Tracking). BFM-Zero demonstrated superior sample efficiency, achieving higher tracking accuracy with only 200M samples compared to GMT’s 6800M samples. It also showed greater robustness in recovery scenarios where task-specific baselines often fail.
- Quantitative Real-world Validation (Reviewer NnXb): The authors addressed concerns regarding the sim-to-real gap by reporting physical tracking metrics. The average joint position error ($E_{mpjpe}$) on the real robot was recorded at 0.8041 rad, with a minimal discrepancy of only 0.0117 rad per joint compared to simulation.
- Latent Space Quantification (Reviewer qsJd, NnXb): New analyses in the appendix (clustering, Earth Mover's Distance) confirmed that the latent space is semantically organized and captures a diverse range of motions beyond the initial training set
- Ablation of Design Choices (Reviewer qyhf): The authors substantiated the necessity of Domain Randomization (DR) and Auxiliary Rewards through hardware tests. While simulation performance appeared similar without them, physical deployment failed due to excessive jittering or foot slippage, proving their critical role in stability.

Despite the strong consensus for acceptance, a few points remain for consideration:
- Technical Novelty (Reviewer qsJd, NnXb): While the hardware deployment is non-trivial, it was noted that the underlying FB-CPR framework has substantial overlap with prior work. The technical contribution is viewed by some as more of an engineering and scaling achievement than a fundamental algorithmic shift.
- Discriminator Scalability (Reviewer NnXb): Questions persist regarding whether the GAN-style discriminator can scale to even larger, more heterogeneous datasets without experiencing mode collapse.

**Reviewer Scores:**

- Reviewer yhgB (Score 6) : I expect this reviewer to increase the score to 8
- Reviewer qsJd (Score 4) : I expect this reviewer to increase the score to 6
- Reviewer NnXb (Score 6) : I expect this reviewer to increase the score to 8
- Review qyhf (Score 6) : I expect this review to increase the score to 8

---

### Decision · Program_Chairs · 2026-01-26

Accept (Poster)